# Structural plasticity of mumps virus nucleocapsids with cryo-EM structures

Hong Shan[1,2,7], Xin Su[1,7], Tianhao Li[1,7], Yuqi Qin[1,7], Na Zhang[1,3], Liuyan Yang[4,5], Linsha Ma[6], Yun Bai[6], Lei Qi[2], Yunhui Liu[1] & Qing-Tao Shen [1,2 ✉]

Mumps virus (MuV) is a highly contagious human pathogen and frequently causes world-wide outbreaks despite available vaccines. Similar to other mononegaviruses such as Ebola and rabies, MuV uses a single-stranded negative-sense RNA as its genome, which is enwrapped by viral nucleoproteins into the helical nucleocapsid. The nucleocapsid acts as a scaffold for genome condensation and as a template for RNA replication and transcription. Conformational changes in the MuV nucleocapsid are required to switch between different activities, but the underlying mechanism remains elusive due to the absence of high-resolution structures. Here, we report two MuV nucleoprotein-RNA rings with 13 and 14 protomers, one stacked-ring filament and two nucleocapsids with distinct helical pitches, in dense and hyperdense states, at near-atomic resolutions using cryo-electron microscopy. Structural analysis of these in vitro assemblies indicates that the C-terminal tail of MuV nucleoprotein likely regulates the assembly of helical nucleocapsids, and the C-terminal arm may be relevant for the transition between the dense and hyperdense states of helical nucleocapsids. Our results provide the molecular mechanism for structural plasticity among different MuV nucleocapsids and create a possible link between structural plasticity and genome condensation.

[1] iHuman Institute and School of Life Science and Technology, ShanghaiTech University, Shanghai, China. [2] Laboratory for Marine Biology and Biotechnology, Qingdao National Laboratory for Marine Science and Technology, Qingdao, China. [3] University of Chinese Academy of Sciences, Beijing, China. [4] State Key Laboratory of Microbial Technology, Marine Biotechnology Research Center, Shandong University, Qingdao, China. [5] College of Marine Life Sciences, Ocean University of China, Qingdao, China. [6] School of Life Science and Technology, ShanghaiTech University, Shanghai, China. [7] These authors contributed equally: Hong Shan, Xin Su, Tianhao Li, Yuqi Qin. ✉email: shenqt@shanghaitech.edu.cn

The mumps virus (MuV) is a member of the order *Mononegavirales* and causes a contagious disease with symptoms ranging from parotitis to mild meningitis and severe encephalitis[1–3]. Despite available vaccines, MuV still causes infections in many regions of the world and even in people with a history of vaccination. No antiviral treatments are yet available[4]. Similar to other mononegaviruses such as Ebola, rabies, and Newcastle disease viruses (NDV), MuV uses a single-stranded negative-sense RNA as its genome, which is nuclease sensitive. To maintain the integrity of its viral genome, MuV utilizes its nucleoproteins (N) to enwrap the viral genome into a helical nucleocapsid (NC) in which the RNA genome is buried in the cleft between N-terminal domain (NTD) and C-terminal domain (CTD) via electrostatic interaction[5–9].

Measles virus (MeV) and the Sendai virus (SeV) have several tubular nucleocapsid segments that are connected with thin and loose nucleocapsid loops, as revealed by cryo-electron tomography (cryo-ET)[10–12]. Further high-resolution structural analyses on recombinant nucleoproteins and the enwrapped RNA (N-RNA) via either X-ray crystallography or electron microscopy (EM) indicate that the number of protomers per turn varies from 9 to 14 in different mononegaviruses[9,13–24]. Conversely, pitches in the well-resolved helical nucleocapsids including MeV, SeV, and NDV stay in a tight range between 5.0 and 6.8 nm. This causes the helical nucleocapsids to assume a dense state providing protection to the RNA genomes[13,15]. In Ebola virus, such dense nucleocapsids also play vital roles in RNA genome condensation during the maturation of virions[6–8]. During viral genome replication and transcription, RNA-dependent RNA polymerase and its cofactor phosphoprotein (P) will relax these dense nucleocapsids into partially loosened nucleocapsids to enable access[25,26]. Thus, the structural plasticity of mononegaviral nucleocapsids in different states might correlate with genome protection, genome condensation, and genome replication/transcription.

MuV nucleoproteins are common in mononegaviruses. They have been shown to assemble into rings or helices under variable conditions in preliminary biochemical and EM studies[25,27,28]. Specifically, recombinant full-length MuV nucleoproteins enwrap RNA into a nucleocapsid-like ring, which has a diameter of ~20 nm, with 13 protomers[27]. After either trypsin treatment or long-term incubation at 4 °C, MuV nucleoproteins are truncated after residue 379 and exist as a nucleocapsid-like helical structure[28,29]. Further cryo-EM analysis on authentic MuV nucleocapsids along with CTD of phosphoprotein reveals a relatively stable helical structure with a resolution of 18 Å, while the NTD of phosphoprotein plays a distinctive uncoiling role to MuV nucleocapsids[25]. These results indicate the occurrence of different assemblies of MuV nucleocapsids and provide clues on the structural alteration upon binding to phosphoprotein.

Limited by the absence of high-resolution structures, detailed insights on the ability of MuV nucleoprotein to switch between different forms, and their corresponding function, remain elusive. In this work, we heterogeneously expressed MuV nucleoproteins and used cryo-EM as the major approach to resolve 5 high-resolution MuV N-RNA assemblies, including 2 ring-like structures in 13 and 14 protomers, 1 stacked-ring filament, and 2 nucleocapsids with distinct helical pitches. Based on these high-resolution structures, we have clarified the molecular mechanism for structural plasticity among different forms of MuV nucleocapsids and built a possible link between structural plasticity and genome condensation.

## Results

**MuV N-RNA rings in different protomers**. Following the previous protocols[27], MuV nucleoproteins were expressed in an

*Escherichia coli* system and purified using tandem affinity and gel filtration chromatography. Compared with our previous purification of NDV and SeV nucleoproteins, which showed the occurrence of broad peaks[13], MuV nucleoproteins have a sharp peak in the gel filtration profile indicating a homogeneous state (Fig. 1a). Consistent with this, MuV nucleoproteins exhibit uniform ring-like structures under cryo-EM with a diameter of ~19 nm (Fig. 1b) similar to prior work[27]. Direct two-dimensional (2D) classification on vitrified MuV ring-like structures could distinguish the numbers of protomers in each class: ~83% were ring-like particles with 13 protomers and ~17% particles had 14 protomers (Fig. 1b).

Only top-on views were visible in MuV 13- and 14-protomer rings, indicating that strong preferred orientation was present. To fill in the missing cones, 20° and 40° tilted micrographs—together with untilted micrographs—were collected on MuV nucleoproteins for single particle analysis (Table 1 and Supplementary Fig. 1). After 2D and three-dimensional (3D) classifications, 13-protomer and 14-protomer rings of MuV nucleoproteins were separated for further 3D refinements with the enforced 13-fold or 14-fold symmetry. MuV 13- and 14-protomer rings were finally resolved at the respective resolutions of 3.3 and 6.2 Å (denoted as $N_{ring-13p}$ and $N_{ring-14p}$, respectively), based on 3D Fourier shell correlation (FSC; Supplementary Fig. 2). Considering that the atomic model for MuV nucleoprotein was still undetermined, homology modeling on MuV nucleoprotein based on the 64% sequence similarity of nucleoproteins between MuV and parainfluenza virus 5 (PIV5) was used and resulted in a model for MuV protomer. Copies of this atomic model were flexibly docked into the EM map and optimized for better local density fitting in $N_{ring-13p}$ and $N_{ring-14p}$ (Fig. 1c, d).

Similar to nucleoprotein structures from nearly all mononegaviruses[9,14,15], the C-terminal tail (N-tail, residues from 407 to 549) of MuV nucleoprotein is invisible in the EM maps of $N_{ring-13p}$ and $N_{ring-14p}$ due to the intrinsic flexibility. To our surprise, the C-arm (residues from 374 to 406), which is resolved in many mononegaviruses such as the rabies virus, vesicular stomatitis virus (VSV), respiratory syncytial virus (RSV), PIV5, and MeV[9,14–16,18], was not determined in either MuV $N_{ring-13p}$ or $N_{ring-14p}$ (Supplementary Figs. 3 and 4). We speculate that the C-arms of MuV $N_{ring-13p}$ and $N_{ring-14p}$ are flexible, as the N-tails, and are averaged out during the 3D reconstruction. Thus, the flexible C-arm of MuV $N_{ring-13p}$ or $N_{ring-14p}$ will not form a stable interface with the α16 helix from the neighboring protomer. This might not play an essential role in nucleoprotein oligomerization as in MeV, NDV, and PIV5[13,15,16,30].

Without the assistance of C-arms, N-arms from MuV $N_{ring-13p}$ and $N_{ring-14p}$ play a central role in holding neighboring protomers together via domain swapping. Specifically, the N-arm from $N_i$ interacts with both α12 helix and the N-arm from $N_{i+1}$ to form a stable helix bundle (Fig. 1e, f). In addition to the N-arm domain swapping interface, there is another interface between neighboring protomers that has not been noticed previously: MuV nucleoprotein has an extended loop ($Loop_{20-46}$) connecting the N-arm and the core of NTD. The $Loop_{20-46}$, along with a loop from CTD ($Loop_{312-320}$) and a loop from the NTD ($Loop_{92-102}$) assembles into a hole, which is adjacent to the N-arm (denoted as the N-hole). Another loop ($Loop_{242-250}$), extending from the NTD of $N_{i-1}$ can become inserted into the N-hole from $N_i$. The electrostatic interaction between $Loop_{242-250}$ (positively charged) and the N-hole (negatively charged) tightly anchors neighboring nucleoproteins into the proper positions (Fig. 1e–g). Therefore, the N-hole adopts the same domain swapping process as the N-arm and contributes to the assembly of MuV $N_{ring-13p}$ and $N_{ring-14p}$. Recently, similar N-hole-like motifs have also been reported to stabilize SeV nucleocapsids[31].

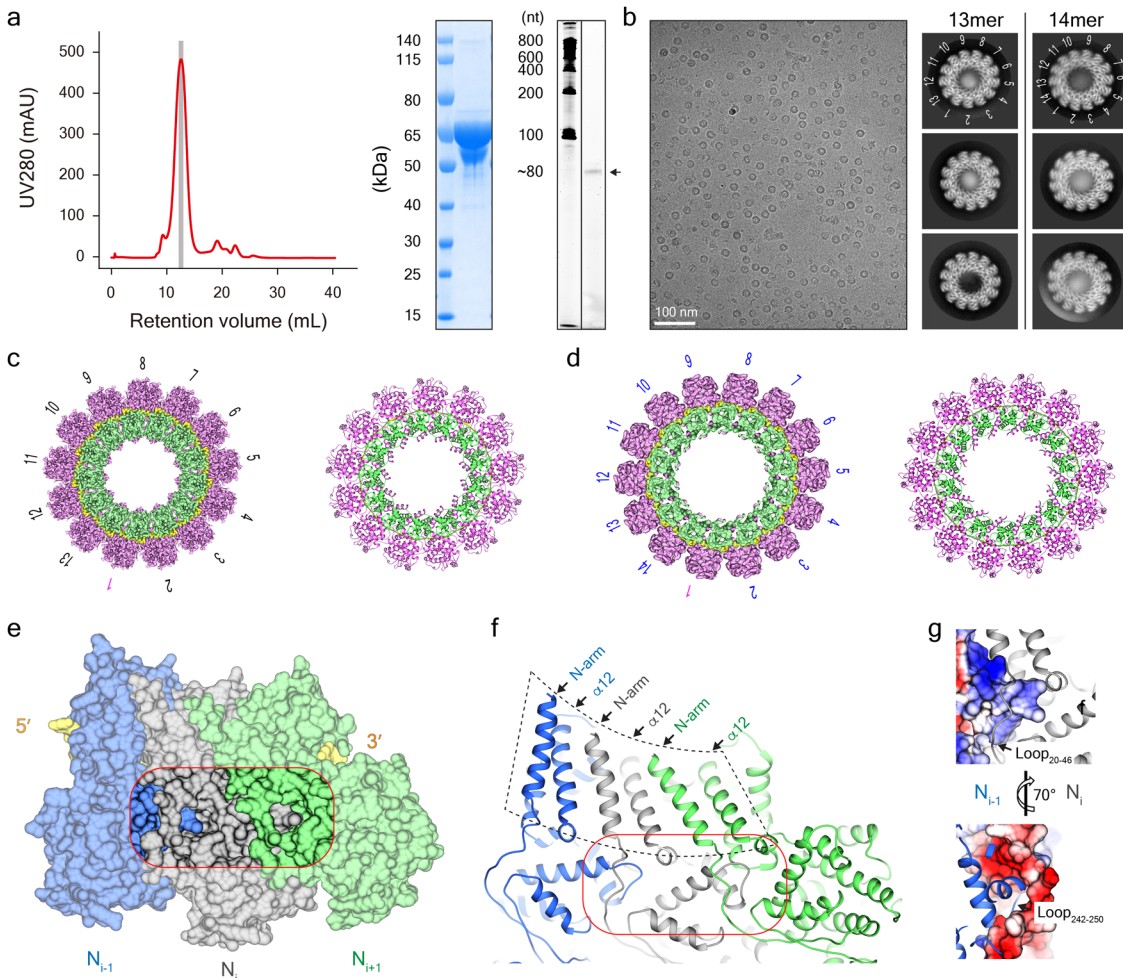

**Fig. 1 MuV N-RNA rings in different protomers. a** Gel filtration chromatography of MuV nucleoproteins (left) and the respective analysis of SDS-PAGE gel (middle) and RNA gel (right). **b** A typical cryo-EM image of MuV N-RNA rings (left) and the 2D classification average galleries (right) of $N_{ring-13p}$ (~83% particles) and $N_{ring-14p}$ (~17% particles). **c** 3D reconstruction of MuV $N_{ring-13p}$ and the respective atomic model. RNA, NTD, and CTD are colored in yellow, pink and green, respectively. The same color code is used for the rest of the figures unless specified. **d** 3D reconstruction of MuV $N_{ring-14p}$ and the respective atomic model. **e**, **f** Domain swapping and the detailed structural analysis on the N-arm and N-hole in MuV N-RNA rings. **g** The electrostatic interaction between $Loop_{242-250}$ from $N_{i-1}$ and the N-hole from $N_i$.

**Structural plasticity of MuV N-RNA rings.** Previous structural studies have shown that the number of protomers per turn in a given mononegaviral nucleoprotein is usually fixed. Only nucleoproteins from different mononegaviruses have different numbers of protomers varying from 9 to 14[9,13–15]. The co-existence of 13- and 14-protomer rings indicates the structural plasticity inherent in MuV nucleoproteins and provides an opportunity to investigate its underlying assembly mechanism. Compared with $N_{ring-13p}$, $N_{ring-14p}$ has one more protomer, and the corresponding twist angle of $N_{ring-14p}$ is 25.7°, reduced from 27.7° in $N_{ring-13p}$ (Table 2).

After one protomer (labeled as 1st) from $N_{ring-13p}$ was merged with one from $N_{ring-14p}$, all protomers from $N_{ring-13p}$ share the same plane with protomers of $N_{ring-14p}$ (Fig. 2a). The transition from MuV $N_{ring-13p}$ to $N_{ring-14p}$ comes from the in-plane rotation of each protomer. Indeed, the 2nd protomer of $N_{ring-13p}$ rotates ~2.0° around the residue $P_{213}$ and reaches the position for the 2nd protomer in $N_{ring-14p}$. As a result, the rotation distance between residue $M_1$ (furthermost from the rotation center) in $N_{ring-13p}$ and $N_{ring-14p}$ reaches 2.0 Å. The accumulated change after 12 iterations yields an ~50 nm³ space between the 1st protomer and the 13th protomer in $N_{ring-13p}$, which allows for the 14th protomer from $N_{ring-14p}$. Accordingly,

the diameter of $N_{ring-14p}$ increases 1.4 nm to 20.3 nm relative to $N_{ring-13p}$ (Fig. 2a, b).

Along with the structural switch between neighboring protomers in MuV $N_{ring-13p}$ and $N_{ring-14p}$, the interfaces involved by swapping N-arms and N-holes will be influenced. Compared to $N_{ring-13p}$, the α12 helix of $N_{i+1}$ from $N_{ring-14p}$ moves 1.2–1.4 Å further away from N-arm of $N_i$ (Fig. 2c). The same phenomenon occurs at the other interfaces between the N-hole and the bulged $Loop_{242-250}$. A greater distance between the residues at these two interfaces results in a weaker interaction in $N_{ring-14p}$ than in $N_{ring-13p}$, which helps explain why $N_{ring-13p}$ is dominant in MuV N-RNA rings.

**MuV N-RNA filaments in stacked rings.** Purified MuV nucleoproteins have strong predilection to assemble into rings, which only enwrap a limited number of nucleotides and are obviously not biologically relevant (Fig. 1a). The NTD or CTD of the MuV phosphoproteins can directly interact with N-tails of MuV nucleoproteins—this might be related to the fact that authentic MuV nucleoproteins enwrap RNA into either uncoiled or thicker helical nucleocapsids[25]. Following this idea, most MuV nucleoproteins assembled into helices with only a small portion of ring-like structures when N-tails were removed via trypsin digestion or deletion mutation (Supplementary Fig. 5).

**Table 1 Cryo-EM data collection and data processing statistics.**

| | $N_{ring-13p}$ (EMDB-31361) (PDB-7EWQ) | $N_{ring-14p}$ (EMDB-30281) (PDB-7EWQ) | $NC_{helix-dense}$ (EMDB-31368) (PDB-7EXA) | $NC_{helix-hyper}$ (EMDB-31369) (PDB-7EXA) | $N_{ring-stacked}$ (EMDB-31370) (PDB-7EXA) | $NC_{helix-\Delta arm}$ (EMDB-31367) (PDB-7EXA) |
|---|---|---|---|---|---|---|
| *Data collection and processing* | | | | | | |
| Microscope | Titan Krios G$^{3i}$ | Titan Krios G$^{3i}$ | Titan Krios G$^2$ | Titan Krios G$^2$ | Titan Krios G$^2$ | Titan Krios G$^2$ |
| Voltage (kV) | 300 | 300 | 300 | 300 | 300 | 300 |
| Camera | Gatan K3 Bio-Quantum | Gatan K3 Bio-Quantum | Gatan K2 Summit | Gatan K2 Summit | Gatan K2 Summit | Gatan K3 Summit |
| Magnification | 81,000 | 81,000 | 18,000 | 18,000 | 18,000 | 18,000 |
| Electron exposure (e$^-$/Å$^2$) | 50 | 50 | 40 | 40 | 40 | 40 |
| Defocus range (μm) | 1.5–3 | 1.5–3 | 1.5–3 | 1.5–3 | 1.5–3 | 1.5–3 |
| Pixel size (Å) | 0.53 | 0.53 | 0.65 | 0.65 | 0.65 | 0.66 |
| Symmetry imposed | C13 | C14 | Helical | Helical | C13 | Helical |
| Initial particle images (no.) | 1,012,258 | 1,012,258 | 574,262 | 574,262 | 574,262 | 242,861 |
| Final particle images (no.) | 390,418 | 84,124 | 45,506 | 38,110 | 79,901 | 192,387 |
| Map resolution (Å) | 3.3 | 6.2 | 3.9 | 3.6 | 3.7 | 2.9 |
| FSC threshold | 0.143 | 0.143 | 0.143 | 0.143 | 0.143 | 0.143 |
| Map resolution range (Å) | 3.3–10 | 6.2–15 | 3.6–4.2 | 3.4–3.8 | 3.5–3.9 | 2.4–3 |
| *Refinement* | | | | | | |
| Initial model used (PDB code) | 4XJN | 4XJN | 6JC3 | 4UFT | 6JC3 | — |
| Model resolution (Å) | 3.3 | 6.2 | 3.9 | 3.6 | 3.7 | 2.9 |
| FSC threshold | 0.143 | 0.143 | 0.143 | 0.143 | 0.143 | 0.143 |
| Map sharpening *B* factor (Å$^2$) | −135.99 | −135.99 | −188.62 | −157.92 | −173.72 | −127.25 |
| Model composition | | | | | | |
| Non-hydrogen atoms | 3088 | 3088 | 2968 | 2968 | 2968 | 2968 |
| Protein residues | 374 | 374 | 374 | 374 | 374 | 374 |
| Ligands | 0 | 0 | 0 | 0 | 0 | 0 |
| *B* factors (Å$^2$) | | | | | | |
| Protein | 164.92 | 164.92 | 15.57 | 15.57 | 15.57 | 15.57 |
| Ligand | — | — | — | — | — | — |
| R.m.s. deviations | | | | | | |
| Bond lengths (Å) | 0.011 | 0.011 | 0.011 | 0.011 | 0.011 | 0.011 |
| Bond angles (°) | 1.161 | 1.161 | 1.161 | 1.161 | 1.161 | 1.161 |
| Validation | | | | | | |
| MolProbity score | 2.28 | 2.28 | 1.78 | 1.78 | 1.78 | 1.78 |
| Clashscore | 22.88 | 22.88 | 11.79 | 11.79 | 11.79 | 11.79 |
| Poor rotamers (%) | 0.32 | 0.32 | 1.6 | 1.6 | 1.6 | 1.6 |
| Ramachandran plot | | | | | | |
| Favored (%) | 93.55 | 93.55 | 97.85 | 97.85 | 97.85 | 97.85 |
| Allowed (%) | 6.18 | 6.18 | 2.15 | 2.15 | 2.15 | 2.15 |
| Disallowed (%) | 0.27 | 0.27 | 0 | 0 | 0 | 0 |

To determine whether a partial N-tail can enable such transitions, freshly purified full-length MuV nucleoproteins were kept at 4 °C for 4 weeks, as previously described[28]. Due to the susceptibility to digestion by residual impurities, aged MuV nucleoprotein showed several bands from ~45 to ~55 kDa (Fig. 3a). Interestingly, aged MuV nucleoprotein also reassembled into long and straight filaments under cryo-EM (Fig. 3b), which were chosen for the following structural analysis.

MuV nucleocapsid filaments were segmented with 90% overlaps for helical reconstruction. After the 2D classification, particles could be grouped into three typical classes: ~30% particles show parallel layers with the interval at ~4.5 nm, ~47% particles form helices with the helical pitch at ~5.6 nm, and ~23% particles assemble into even denser helices with a helical pitch at ~4.5 nm (Fig. 3c). These three kinds of structures were reconstructed separately with different helical parameters applied.

Parallel layers of MuV N-RNA filaments were reconstructed at 3.7-Å resolution based on gold-standard FSC; the reconstruction comprises layers of rings packed in a head-to-tail mode (denoted as $N_{ring-stacked}$), which is distinct from VSV N-RNA double rings in a head-to-head manner in the crystallographic unit cell[9]. Each layer of MuV $N_{ring-stacked}$ has 13 protomers as $N_{ring-13p}$ (Fig. 3d

and Supplementary Fig. 6), and one atomic protomer structure from $N_{ring-13p}$ was flexibly docked into the map of $N_{ring-stacked}$ optimized for better local density fitting and then duplicated 12 times to build the atomic model for one layer of $N_{ring-stacked}$ (Fig. 3d). The overall shape between $N_{ring-13p}$ and one layer of $N_{ring-stacked}$ is very similar with a root-mean-square deviation at 0.84 Å. Lateral interactions between neighboring layers might influence the detailed conformation of the $N_{ring-stacked}$, with several loops and the N-arm being poorly aligned (Fig. 3e).

The atomic model of one layer was then copied and docked in the neighboring layers of MuV $N_{ring-stacked}$. Structural analysis on MuV $N_{ring-stacked}$ shows that protomers from the upper layer are not immediately over the neighboring protomers from the lower layer and the twist angle between neighboring layers is 9.3° (Fig. 3f). Such an arrangement lets one protomer from the upper layer slide into the gap between two neighboring protomers in the lower layer and reduces the distance between neighboring layers to 4.5 nm, which is less than the helical pitches at ~5.0 nm in the well-resolved helical MeV nucleocapsids[15,30]. The close contact between neighboring layers renders positively charged α16 helices from $N_i$ and negatively charged Loop$_{20-46}$ from the upper $N_i'$ to form a stable interface (Fig. 3g).

**Table 2 Structural parameters of nucleocapsids in the order of *Mononegavirales*.**

| Viruses | Nucleoproteins | Approach | Oligomeric states | Pitch (nm) | Twist | Resolution (Å) | EMDB \| PDB |
|---|---|---|---|---|---|---|---|
| MuV | Recombinant | Cryo-EM | Ring | n/a | −27.7° | 3.3 | 31361 \| 7EWQ |
| | | | | n/a | −25.7° | 6.2 | 30281 \| 7EWQ |
| | | | Helix | 5.3 | −27.1° | 3.9 | 31368 \| 7EXA |
| | | | | 4.6 | −26.8° | 3.6 | 31369 \| 7EXA |
| | | | | n/a | −27.7° | 3.7 | 31370 \| 7EXA |
| | Authentic | Cryo-EM | Helix | 6.7 | −28.3° | 18.1 | 2630 \| n/a |
| NDV | Recombinant | Cryo-EM | Clam | 5.1 | −27.5° | 4.8 | 9793 \| 6JC3 |
| SeV | Recombinant | Negative stain EM | Helix | 5.3 6.8 37.5 | n/a | n/a | n/a \| n/a |
| | | Cryo-EM | Helix | 5.4 | −27.5° | 4.1 | 30066 \| 6M7D |
| | | | | 5.6 | −27.4° | 4.6 | 30065 \| 6M7D |
| | | | | 5.3 | −27.6° | 2.9 | 30129 \| 6M7D |
| | | | Clam | 5.6 | −27.1° | 3.9 | 30064 \| 6M7D |
| NiV | Recombinant | X-ray | C1 | n/a | n/a | 2.5 | n/a \| 4CO6 |
| | | Cryo-EM | Clam | n/a | −27.9° | 4.3 | n/a \| n/a |
| PIV5 | Recombinant | X-ray | Ring | n/a | −27.7° | 3.1 | n/a \| 4XJN |
| RSV | Recombinant | X-ray | Ring | n/a | −36.0° | 3.3 | n/a \| 2WJ8 |
| MeV | Recombinant | Cryo-negative stain EM | Helix | 5 to 6.6 | −26.8° to −27.6° | n/a | n/a \| n/a |
| | | Cryo-EM | Helix | 4.9 | −29.2° | 4.3 | 2867 \| 4UFT |
| | | X-ray | Ring | n/a | n/a | 2.7 | n/a \| 5E4V |
| | Authentic | Cryo-ET | Helix | 6.4 | n/a | n/a | 1973 \| n/a |

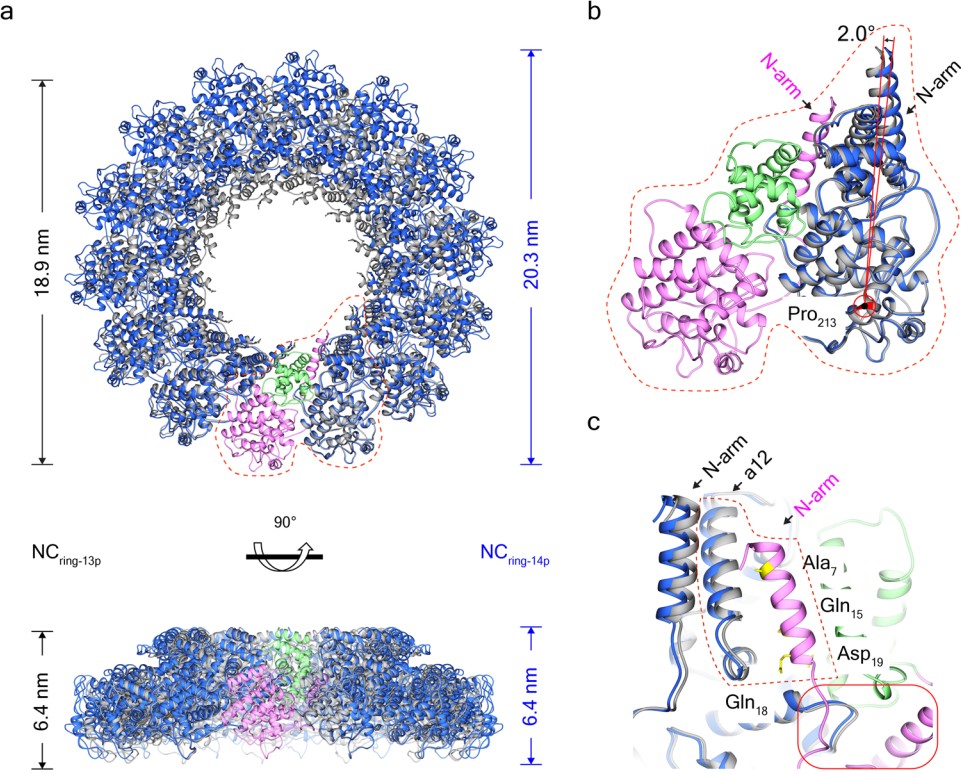

**Fig. 2 Structural plasticity of MuV N-RNA rings. a** Structural comparison between the superimposed MuV $N_{ring-13p}$ and $N_{ring-14p}$. Except the superimposed protomers (the 1st, colored in pink), the other protomers in $N_{ring-13p}$ and $N_{ring-14p}$ are colored in gray and blue, respectively. Side views of $N_{ring-13p}$ and $N_{ring-14p}$ have the same height at 6.4 nm and share the same plane. **b** In-plane rotation of the 2nd protomer in the superimposed MuV $N_{ring-13p}$ and $N_{ring-14p}$. **c** Slight interface change of N-arm and N-hole in the superimposed MuV $N_{ring-13p}$ and $N_{ring-14p}$. The N-arm and the N-hole interfaces between neighboring protomers are marked in dashed and solid polygons, respectively.

**Dense and hyperdense MuV helical nucleocapsids.** Besides the MuV $N_{ring-stacked}$, there are two other kinds of helical filaments in different pitches based on the 2D classification (Fig. 3c). The helical filament with a helical pitch at ~5.6 nm was reconstructed with a final resolution of 3.9 Å (Fig. 4a and Supplementary Fig. 6). Compared with the uncoiled MuV nucleocapsid in association with the NTD of phosphoprotein[25], this straight helical filament was termed the dense nucleocapsid (denoted as $NC_{helix-dense}$). $NC_{helix-dense}$ is left-handed with a twist angle at −27.1°; accordingly, the number of protomers per turn is 13.3 between $N_{ring-13p}$ and $N_{ring-14p}$. The rise of $NC_{helix-dense}$

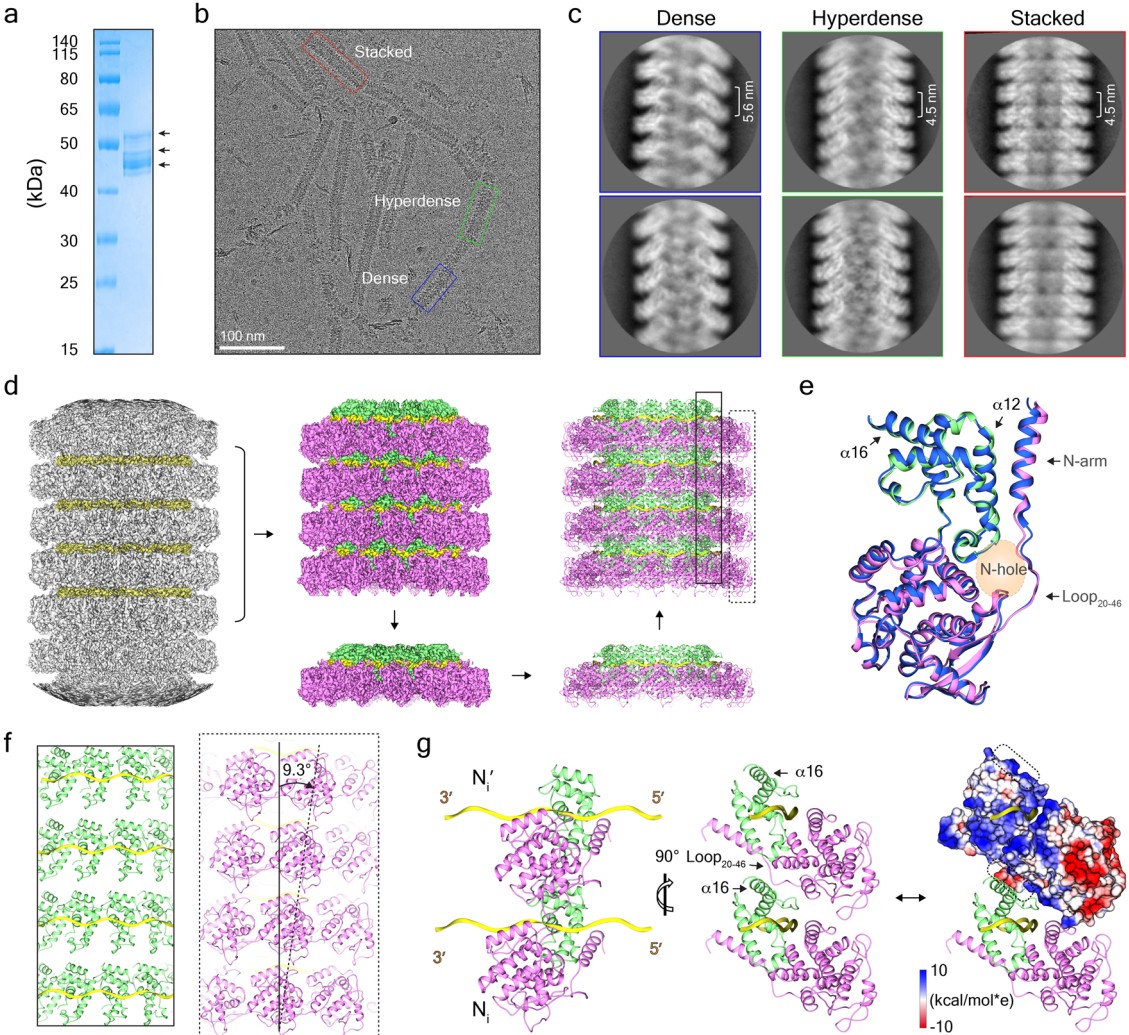

**Fig. 3 MuV N-RNA filament in stacked rings. a** The SDS-PAGE gel of aged MuV nucleoproteins (kept under 4 °C for 4 weeks). **b** A representative cryo-EM image of aged MuV nucleoproteins. Dense, hyperdense, and ring-stacked filaments are highlighted in blue, green, and red boxes, respectively. **c** Typical 2D classification average of dense, hyperdense, and ring-stacked filament segments. The respective helical pitches are marked. **d** 3D reconstruction of $N_{ring\text{-}stacked}$ and the respective atomic models. **e** Structural comparison between protomers from $N_{ring\text{-}13p}$ and $N_{ring\text{-}stacked}$. The $N_{ring\text{-}13p}$ protomer is colored in blue. **f** The twist of protomers between neighboring layers of MuV $N_{ring\text{-}stacked}$. Two sliced views of neighboring layers are shown as in **d** with the twist angle marked. **g** Interface analysis of protomers between neighboring layers of MuV $N_{ring\text{-}stacked}$. The electrostatic distribution on $Loop_{20\text{-}46}$ and α16 helix from $N_i'$ is marked.

is 4.2 Å, and the helical pitch is 5.6 nm, which nicely fits our measurements in 2D classification. The atomic model of one nucleoprotein from $N_{ring\text{-}stacked}$ was docked well into the EM density of $NC_{helix\text{-}dense}$ in a rigid body (Fig. 4b and Supplementary Fig. 6). Besides the well-suited residues from 1 to 373, the extra density in $NC_{helix\text{-}dense}$ was described as the C-arm and is unresolved in $N_{ring\text{-}13p}$, $N_{ring\text{-}14p}$, or $N_{ring\text{-}stacked}$.

The other MuV nucleocapsid with a smaller helical pitch at ~4.5 nm was reconstructed at a final resolution of 3.6 Å (Fig. 4c and Supplementary Fig. 6). This helical nucleocapsid is also left-handed and contains a twist angle of −26.8° and 13.4 protomers per turn. The rise of this helix is 3.4 Å, which is much smaller than that of $NC_{helix\text{-}dense}$. We described this type of helix as a hyperdense MuV nucleocapsid (denoted as $NC_{helix\text{-}hyper}$). The atomic model of one nucleoprotein from $NC_{helix\text{-}dense}$ was fitted into the EM density of $NC_{helix\text{-}hyper}$ in a rigid body, and only residues from 1 to 373 were clearly resolved (Fig. 4d and Supplementary Fig. 6). The C-arm is missing in the $NC_{helix\text{-}hyper}$ similar to $N_{ring\text{-}13p}$, $N_{ring\text{-}14p}$, and $N_{ring\text{-}stacked}$.

To double check whether $NC_{helix\text{-}hyper}$ was formed due to the absence of C-arm, we designed a truncation mutation on MuV nucleoprotein with both the C-arm and the N-tail removed and obtained the pure protein for structural analysis; the truncation mutation assembled into long and hyperdense nucleocapsids (denoted as $NC_{helix\text{-}\Delta arm}$), which were then examined through a 3D reconstruction (Supplementary Fig. 7). $NC_{helix\text{-}\Delta arm}$ was reconstructed at a resolution at 2.9 Å with almost the same helical rise and twist as $NC_{helix\text{-}hyper}$. The well-resolved cryo-EM map helps build an accurate atomic model for MuV $NC_{helix\text{-}\Delta arm}$, which is used to polish the other atomic models obtained above (Supplementary Figs. 7 and 8).

In either MuV $NC_{helix\text{-}hyper}$ or $NC_{helix\text{-}\Delta arm}$, superimposition of the C-arm on the protomer $N_i$ will cause a clash with $Loop_{20\text{-}46}$ of $N_{i+14}$ in the upper rung (Fig. 4e). In $NC_{helix\text{-}dense}$, the C-arm, lying above the α16 helix contacts $Loop_{20\text{-}46}$ via electrostatic interaction and functions like a lift to raise the upper rung up. The absence of the C-arm from $NC_{helix\text{-}dense}$ will expose α16 helix to the $Loop_{20\text{-}46}$ from the upper rung forming the same interface

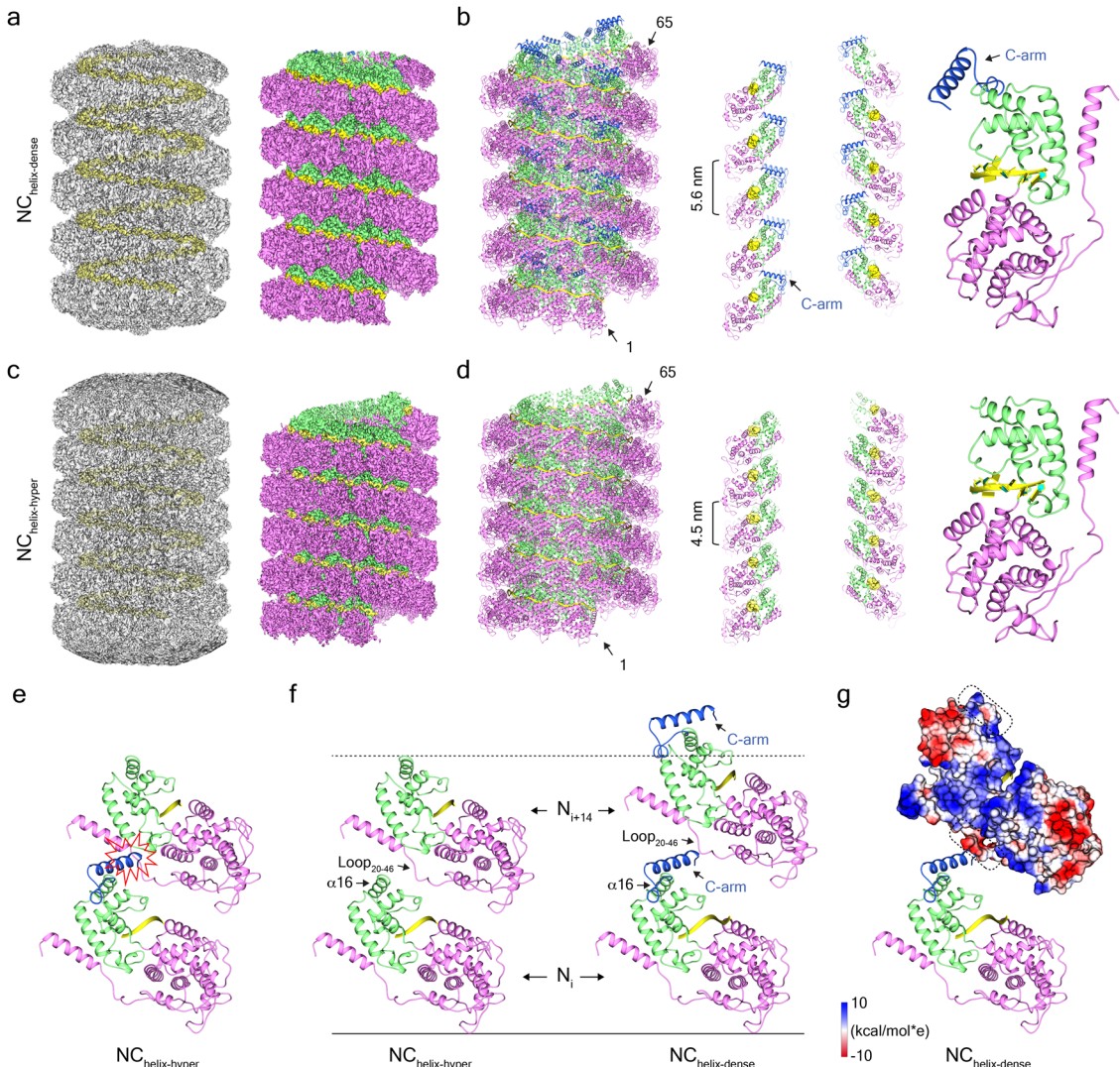

**Fig. 4 Dense and hyperdense MuV helical nucleocapsids. a, b** 3D reconstruction of $NC_{helix-dense}$ and the respective atomic model. The C-arm (residues from 374 to 406) is labeled in blue. Sixty-five protomers in $NC_{helix-dense}$ are shown in **b**. **c, d** 3D reconstruction of $NC_{helix-hyper}$ and the respective atomic model. Sixty-five protomers in $NC_{helix-hyper}$ are shown in **d**. **e** Structural clash in $NC_{helix-hyper}$ with the superimposed C-arm. **f** Interface analysis of protomers from neighboring rungs in $NC_{helix-dense}$ and $NC_{helix-hyper}$. **g** The electrostatic distribution analysis on $N_{i+14}$ in $NC_{helix-dense}$. $Loop_{20-46}$ and $\alpha16$ helix from $N_{i+14}$ are marked with rectangular shapes.

as in $N_{ring-stacked}$ and incurring MuV nucleocapsids that assemble into a hyperdense state (Fig. 4f, g). C-arm is immediately followed by the N-tail in sequence and structure, and the interaction between the N-tail and MuV phosphoprotein might relocate C-arm; the C-arm is speculated to be highly relevant to the assembly of MuV nucleoproteins into either $NC_{helix-dense}$ or $NC_{helix-hyper}$ in different locations.

**Structural plasticity of MuV helical nucleocapsids**. The high-resolution structures of MuV N-RNA rings and helical nucleo-capsids offer the opportunity to study the molecular mechanism for structural plasticity. After one protomer from MuV $NC_{helix-dense}$ was superimposed with one protomer from $N_{ring-stack}$, the helical axis of $NC_{helix-dense}$ tilted anticlockwise at ~8.0° relative to the helical axis of $N_{ring-stacked}$. By contrast, each protomer from MuV $NC_{helix-dense}$ will take a clockwise tilt at ~8.0° to guarantee the parallel helical axes between $NC_{helix-dense}$ and $N_{ring-stacked}$ (Fig. 5a–c and Supplementary Movie 1). Similar tilt transition will occur between $NC_{helix-hyper}$ and $N_{ring-stacked}$; the tilting angle is ~6.5°.

Deduced from the above, the relative tilting angle of each protomer between $NC_{helix-dense}$ and $NC_{helix-hyper}$ is ~1.5°. In all transition processes, the rotation of each protomer is centered around residue $D_{263}$, which is located inside the RNA cleft between NTD and CTD. Notably, the respective distance between the residue $D_{263}$ in the neighboring protomers in $NC_{helix-dense}$ and $NC_{helix-hyper}$ is 26.7 and 26.5 Å, which is very close to the value in $N_{ring-stacked}$ at 26.5 Å (Fig. 5c). Thus, RNA strands between neighboring protomers will maintain very similar lengths regardless of the different helical pitches in MuV $NC_{helix-dense}$, $NC_{helix-hyper}$, and $N_{ring-stacked}$ (helical pitch at 0 nm).

In MuV $N_{ring-13p}$ and $N_{ring-14p}$, endogenous RNAs from E. coli were identified in the EM maps, and poly-Uracils (poly-Us) were modeled into the EM maps to mimic the cellular RNAs. As with other mononegaviral nucleocapsids[14–16,18], the RNA was deeply buried in the RNA cleft between NTD and CTD of MuV nucleoproteins following the "rule of six" with alternating three-base-in and three-base-out conformations. There are 78 and 84 nucleotides in $N_{ring-13p}$ and $N_{ring-14p}$, respectively, which fit well with the band at ~80 nucleotides in the RNA gel on freshly

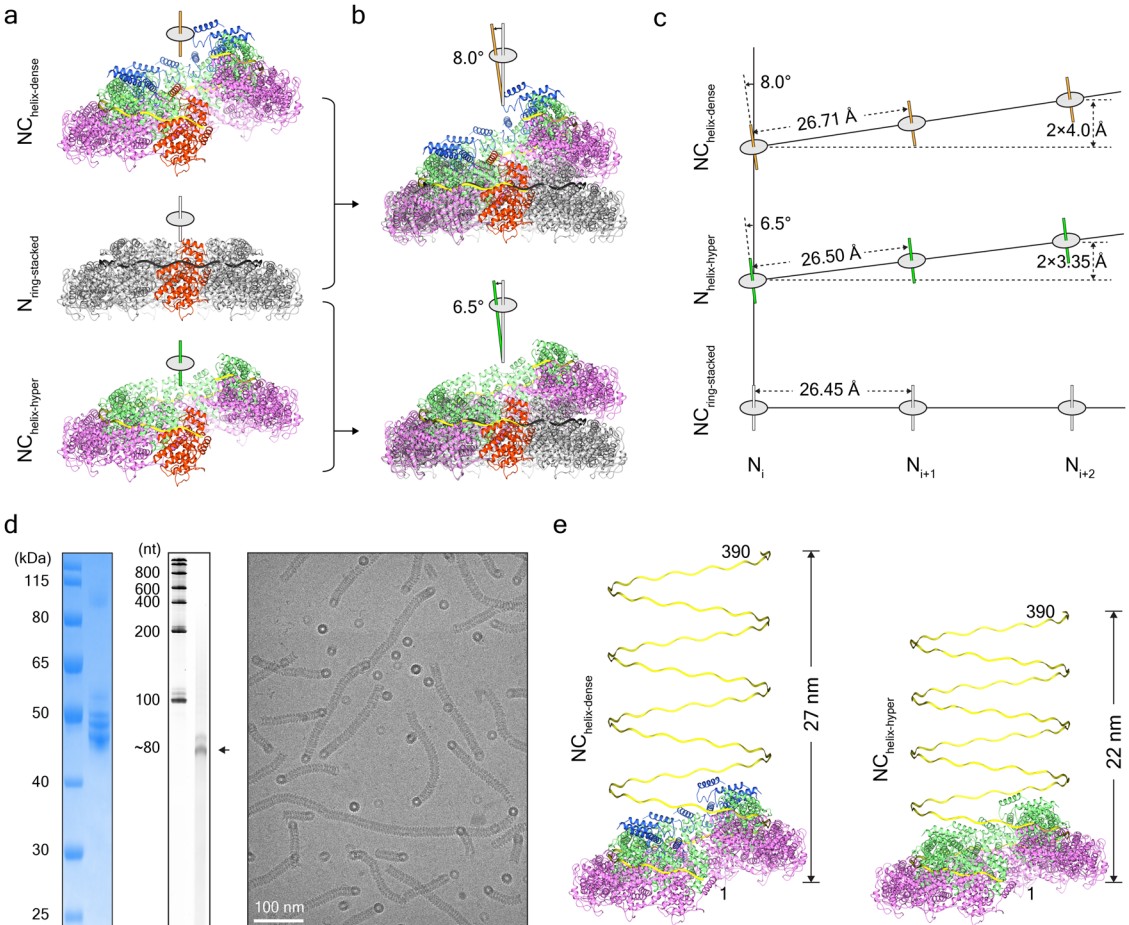

**Fig. 5 Structural plasticity of MuV helical nucleocapsids. a, b** Structural comparison among $N_{ring-stacked}$, $NC_{helix-dense}$, and $NC_{helix-hyper}$. The helical axis for each structure is set upright (**a**). One protomer from $NC_{helix-dense}$ or $NC_{helix-hyper}$ is superimposed with one from $N_{ring-stacked}$, and the angles of helical axes between $NC_{helix-dense}$ and $N_{ring-stacked}$ as well as between $NC_{helix-hyper}$ and $N_{ring-stacked}$ are labeled (**b**). **c** Diagrams of filament assembly of $NC_{helix-dense}$ and $NC_{helix-hyper}$ after the superimposition with $N_{ring-stacked}$. The helical axis for $N_{ring-stacked}$ is set upright. Distance between neighboring protomers and helical rises is marked. **d** The SDS-PAGE gel (left) and RNA gel (middle) of aged MuV nucleoproteins and a representative cryo-EM image of aged MuV nucleoproteins (right). **e** Genome condensation from $NC_{helix-dense}$ to $NC_{helix-hyper}$. The heights of 390-nt RNA in $NC_{helix-dense}$ and $NC_{helix-hyper}$ are marked.

purified MuV nucleoproteins (Fig. 1a). After 4 weeks or even longer (180 days) incubation at 4 °C, full-length MuV nucleoproteins were cleaved after the N-tail or C-arm via the residual impurities, and MuV nucleoproteins assemble into helical nucleocapsids (Fig. 5d). Intriguingly, long-term incubation had no obvious influence on RNA, and clear EM densities for RNAs could still be resolved in MuV $NC_{helix-dense}$, $NC_{helix-hyper}$, and $N_{ring-stacked}$. The RNAs in these MuV filaments follow the exact same assembly pattern as in $N_{ring-13p}$ and $N_{ring-14p}$ (Figs. 3d and 4a, c).

The MuV RNA genome has 15,384 nucleotides. Nucleoprotein plays a key role in the packing of MuV genome before maturation with the help of the matrix protein and phosphoprotein. Following the $NC_{helix-dense}$ pattern, the nucleocapsid enwraps the whole genome and reaches up to 1 μm in length and needs to fold into several segments to fit into a virion (averaged diameter at ~200 nm). Compared with $NC_{helix-dense}$, $NC_{helix-hyper}$ has 15% reduction in helical pitch and can reduce the whole length of helical nucleocapsid to 859 nm, which makes it feasible to pack the whole RNA genome in the tiny space of the virion (Fig. 5e and Supplementary Movie 2). More intriguingly, $NC_{helix-dense}$ and $NC_{helix-hyper}$ could come from the same nucleocapsid filament digested by residual impurities (Fig. 3b) as revealed by tracking the particles belonging to different structures in the same micrograph during 3D reconstruction. Structural compatibility

between $NC_{helix-dense}$ and $NC_{helix-hyper}$ offers MuV nucleocapsids great plasticity for the genome packaging process. Notably, similar structural plasticity and compatibility also occur in MeV nucleocapsids[32].

## Discussion

Ring-like structures of nucleoproteins have been purified and resolved in case of many mononegaviruses, such as MuV, VSV, and RSV, via X-ray crystallography and EM[14,18,33,34]. It is obvious that ring-like structures are not biologically relevant and will likely not be generated during actual viral infections. However, due to the absence of high-resolution structures of different forms, especially from the same species, the molecular mechanism by which nucleoproteins are prevented from oligomerizing into ring-like structures is still missing. A series of high-resolution MuV N-RNA structures hint that the relocation of the N-tail by phosphoprotein or other proteins will not form ring-like structures but will rather cause them to assemble into helical nucleocapsids.

Further structural comparisons between MuV $N_{ring-13p}$ and $N_{ring-14p}$ indicate that the co-existence of $N_{ring-13p}$ and $N_{ring-14p}$ comes from the in-plane rotation (~2.0°) of each protomer around the residue $Pro_{213}$ (Fig. 2b); the switch from MuV ring-like structures to helical nucleocapsids is accompanied with an

out-of-plane tilting (~6.5–8.0°) around the residue $D_{263}$ of each protomer (Fig. 5a–c). These in-plane and out-of-plane rotations for each protomer probably compose the basic movements for MuV structural plasticity.

Relative to both $NC_{helix-dense}$ and $NC_{helix-hyper}$, authentic MuV nucleocapsids along with the CTD of phosphoprotein have a larger twist angle at −28.3° with the number of protomers per turn at 12.7. The larger helical rise is at 6.7 nm[25] and extends the scope of structural plasticity of MuV nucleocapsids. Indeed, such structural plasticity of helical nucleocapsids is quite popular among mononegaviruses as summarized in Table 2. SeV nucleocapsids in several different helical pitches (5.3, 6.8, and 37.5 nm) have been observed under negative stain EM[23]. In MeV, cryo-negative staining of recombinant nucleocapsids on EM shows extensive conformational flexibility with the helical pitch ranging from 5.0 to 6.6 nm. Trypsin digestion on MeV nucleocapsids reduces the helical pitch in the range of 4.6 and 5.2 nm, and the N-tail is proposed to regulate such structural plasticity[32]. Interestingly, cryo-ET analysis on nucleocapsids from the MeV virion shows a helical pitch at ~6.4 nm in line with the above values from recombinant nucleocapsids[10].

In our studies, MuV $NC_{helix-dense}$ and $NC_{helix-hyper}$ with the helical pitch at 5.6 and 4.5 nm are resolved via cryo-EM at the respective resolutions of 3.6 and 3.9 Å. High-resolution reconstructions on MuV nucleocapsids and the structural comparison point out that N-tail and C-arm are involved in the regulation of structural plasticity. The cleavage of N-tail or C-arm from full-length MuV nucleoproteins is apparently biologically irrelevant even though the same strategy is widely used to remove flexible regions in high-resolution structural determination on mononegaviral nucleoproteins. The interactions with viral phosphoproteins or host proteins might move the MuV C-arm or N-tail to a new position and adopt similar conformations as $NC_{helix-dense}$ and $NC_{helix-hyper}$, which is worthy of further investigation on nucleoprotein and phosphoprotein supercomplex.

In summary, we investigated the different assembly forms of MuV N-RNA complexes, obtained their respective near atomic-resolution structures, and clarified the structural plasticity inherent in MuV rings in several protomers and helical nucleocapsids with different pitches. We hypothesize that such structural plasticity in helical nucleocapsids may be required to facilitate genome condensation.

## Methods

**Sequence alignment**. Nucleoprotein sequences including MuV (AAF70388.1; accession numbers were obtained from the NCBI Protein database), PIV5 (YP_138511.1), SeV (NP_056871.1), MeV (NP_056918.1), Hendra virus (NP_047106.1), and NDV (YP_009513194.1) were downloaded from NCBI in FASTA format. Geneious was used to align the sequences[35], and the alignment was displayed via ESPript[36].

**Plasmid constructions**. The nucleoprotein gene of MuV was synthesized by GenScript company (China). The full-length gene (1–549) (denoted as $N_{WT}$) and its derivatives with the N-tail (408–549), as well as C-arm and N-tail (375–549) truncated (denoted as $N_{ΔN-tail}$ and $N_{Δarm}$, respectively) were cloned into pET28b plasmids with 6×His-tag on N-termini for gene expression in *E. coli*. All plasmids were verified via gene sequencing before gene expression.

**Protein expression and purification**. MuV $N_{WT}$, $N_{ΔN-tail}$, and $N_{Δarm}$ expressed in *E. coli* were purified via tandem affinity and gel-filtration chromatography. Specifically, pelleted cells were resuspended in a lysis buffer (20 mM Tris-HCl (pH 7.4), 150 mM NaCl) and disrupted via ultrasonic homogenizers (JNBIO, China). After centrifugation at 47,850 × *g* for 30 min, the supernatant was mixed with the nickel-nitrilotriacetic acid resin in a gravity column at 4 °C for 120 min. The column was washed with 50 mL lysis buffer containing 150 mM imidazole. Target proteins were eluted using the lysis buffer containing 500 mM imidazole. Proteins were concentrated and loaded onto a 24 mL Superose 6 increase 10/300 GL chromatography column (GE Healthcare LifeSciences, USA) pre-equilibrated with the lysis buffer. In all, 0.2 mL fractions were collected, 10 µL of which were

subjected to sodium dodecyl sulfate-polyacrylamide gel electrophoresis (SDS-PAGE) and RNA-gel analysis.

All protein samples were freshly made in the following assays except purified $N_{WT}$, which was stored at 4 °C for 4 weeks or even longer for the digestion by residual impurities. Quantitation of bands in SDS-PAGE gels were analyzed densitometrically by using CLIQS (TotalLab, UK).

**RNA-gel assay**. Protease K was added into fresh or aged MuV $N_{WT}$ proteins with the final concentration at 1 mg/mL. The mixture was incubated at 37 °C for 15 min to digest nucleoproteins. Then, 2 × RNA loading dye was incubated with the mixture at 75 °C for 2 min to denature RNA. The final sample was subjected to a 10% polyacrylamide gel containing 7 M urea and stained with SYBR Gold.

**Trypsin digestion assay**. MuV $N_{WT}$ was treated with trypsin to test the change from ring-like structures to helical nucleocapsids. A 40 µL mixture of $N_{WT}$ (final concentration at 1 mg/mL) with trypsin (final concentration at 0.003 mg/mL) was incubated overnight at 4 °C. Four microliters of fractions were taken out for cryo-EM analysis and SDS-PAGE analysis, respectively.

**Cryo-EM sample preparation and data collection**. Four microliters of samples (~1 mg/mL) were applied to glow-discharged holey grids R2/1 (Quantifoil, Ted Pella, USA) with a thin layer of home-made continuous carbon film. The grids were blotted using a Vitrobot Mark IV (Thermo Fisher Scientific, USA) with 1 s blotting time, force level of 2, and humidity of 100% at 4 °C and then immediately plunged into liquid ethane and stored under liquid nitrogen temperature for future cryo-EM imaging. Cryo-EM grids were examined in the low-dose mode on a Talos L120C TEM (Thermo Fisher Scientific, USA) for screening or instant imaging. Snapshots were taken at a magnification of ×73,000 and a defocus set to about −2 µm, using a Ceta™ 16M camera (Thermo Fisher Scientific, USA).

Data collections on satisfactory grids was performed on three Titan Krios microscopes: Titan Krios $G^2$ TEM (Thermo Fisher Scientific, USA), equipped with a K2 Summit direct electron detector (Gatan, USA), which was used in the super-resolution mode with a pixel size of 0.65 Å; Titan Krios $G^2$ TEM (Thermo Fisher Scientific, USA), equipped with a K3 Summit direct electron detector (Gatan, USA), which was used in the super-resolution mode with a pixel size of 0.66 Å; Titan Krios $G^{3i}$ TEM (Thermo Fisher Scientific, USA), equipped with a K3 BioQuantum direct electron detector (Gatan, USA), which was used in the super-resolution mode with a pixel size of 0.53 Å. Special care was taken to perform a coma-free alignment on the microscopes and detailed data collection conditions are listed in Table 1. Image collection on two Titan Krios $G^2$ scopes and the Titan Krios $G^{3i}$ scope was performed with the SerialEM automated data collection software package[37] and the FEI EPU automated data collection software[38], respectively. Datasets from three Titan Krios scopes were subjected to data analysis, separately.

**Cryo-EM data processing and 3D reconstruction**. Three different datasets were collected, including dispersed particles formed by MuV $N_{WT}$, filaments derived from $N_{WT}$ after 4 weeks digestion at 4 °C, and filaments assembled from $N_{Δarm}$. Different reconstruction strategies including helical reconstruction and single particle analysis were applied to filaments and dispersed particles. The detailed workflows for helical reconstruction and single particle analysis are shown in Supplementary Figs. 1, 6, and 8, respectively.

*Helical reconstruction*. Before image processing, raw frames were aligned and summed with dose weighting under MotionCor2.1[39] and the CTF parameters were determined by CTFFIND-4[40]. Image processing was mainly performed in RELION 3.1[41]. Start and end points of helical filaments were manually specified and particles were extracted with ~90% overlap along the helices. Obvious junks were removed based on 2D classification. For the dataset from $N_{WT}$ after 4 weeks digestion at 4 °C, 2D classes were further separated into 3 different groups ($NC_{helix-dense}$, $NC_{helix-hyper}$ and $N_{ring-stacked}$) for independent helical reconstruction according to the helical pattern and pitch. Based on the respective helical pitch of each group, helical structures of nucleoproteins from NDV[42] and MeV[43], as well as the synthesized ring stacked model from NDV were selected, filtered to 20 Å, and used as the initial models. After another round of 3D classifications on each group to remove heterogeneity, the screened datasets were subjected to 3D auto-refinement and Bayesian polishing with the enforced helical ($NC_{helix-dense}$ and $NC_{helix-hyper}$) or C13 ($N_{ring-stacked}$) symmetry.

For $NC_{helix-Δarm}$, screened particles after 2D classification were subjected to 3D refinement with the structure of $NC_{helix-hyper}$ filtered to 20 Å as the initial model. The reconstruction was improved with Bayesian polish and CTF refinement from 3.4 to 2.9 Å.

All the final reconstructions were filtered and sharpened in RELION post-processing session. The resolutions were determined by gold-standard FSC 0.143. The detailed information on twist, rise, and resolution of MuV helical nucleocapsids is shown in Table 2 and Supplementary Figs. 6 and 8.

*Single particle analysis*. To compensate the missing cone caused by preferred orientation of ring-like structures from MuV $N_{WT}$, micrographs at different tilting

angles (0°, 20°, and 40°) were collected. Tilted particle analysis on MuV $N_{WT}$ were performed sequentially in cryoSPARC v.2.14.2[44] and RELION 3.03[41]. Specifically, raw movie stacks at different tilting angles were aligned and summed with dose weighting under MotionCor2.1[39]. The summed micrographs were imported to cryoSPARC for automatic particle picking and patch CTF estimation. Coordinates of each particle and the respective CTF values were passed into RELION. In RELION, particle sets with the same tilting angle were created and subjected to reference-free 2D classifications, separately. Obvious junks were excluded from each particle set. After the second round of 2D classification on each particle set, classes with different tilting angles were combined according to the number of protomers and yielded two new datasets for further 3D refinements: 13-protomer ring ($N_{ring-13p}$) and 14-protomer ring ($N_{ring-14p}$).

One layer of MuV $N_{ring-stacked}$ with 13 protomers and the synthesized ring-like structures with 14 protomers were low-pass filtered separately to 30 Å as the initial models. The initial tilt angle of each particle in the RELION STAR file was set as the tilting angle of the micrograph where the particle is from. A local search on Euler angles was performed to avoid possible local minima pitfall. 3D maps were obtained after 3D refinements with the enforced 13- or 14-fold symmetry, filtered, and sharpened with RELION post-processing session. 3D FSC with 0.143 criteria was performed on $N_{ring-13p}$ and $N_{ring-14p}$ as described[45].

**Model building and structural analysis**. The homology model of MuV nucleoprotein was initially generated by Modeller[46] using the crystal structure of the PIV5 nucleoprotein (RCSB, PDB-4XJN) as the template[47]. Pseudo-atomic model of MuV nucleoprotein was flexibly docked into the EM density of $N_{ring-13p}$ using Rosetta[48]. The atomic model was further optimized for better local density fitting using Coot[49] and real-space refinement in PHENIX[50]. This refined model from $N_{ring-13p}$ was used as the template and docked into other structures, including MuV $N_{ring-14p}$, $N_{ring-stacked}$, $NC_{helix-dense}$, $NC_{helix-hyper}$, and $NC_{helix-\Delta arm}$. The highest resolution of MuV $NC_{helix-\Delta arm}$ at 2.9 Å yielded a high-fidelity atomic model, and this model was used to guide and modify all the other models for better local density fitting using Coot[49] and real-space refinement in PHENIX[50].

The extra EM densities enwrapped between NTD and CTD in all MuV nucleocapsids were assigned as RNA and were docked using poly-U in Coot due to the unspecific binding of nucleoprotein to RNA[49].

The structural analysis including surface electrostatic distribution and structural superimposition was fulfilled in UCSF Chimera[51].

**Reporting summary**. Further information on research design is available in the Nature Research Reporting Summary linked to this article.

## Data availability
The cryo-EM density maps of MuV nucleocapsids were deposited in Electron Microscopy Data Bank (EMDB, https://www.ebi.ac.uk/pdbe/emdb/) with the accession numbers 31361 ($N_{ring-13p}$), 30281 ($N_{ring-14p}$), 31368 ($NC_{helix-dense}$), 31369 ($NC_{helix-hyper}$), 31370 ($N_{ring-stacked}$), and 31367 ($NC_{helix-\Delta arm}$) and the atom coordinates of single N subunit were deposited in the Protein Data Bank (PDB) with the PDB ID codes 7EWQ and 7EXA. All other data are available in the main text or the supplementary materials or with corresponding author upon reasonable request.

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

## Acknowledgements
We are grateful to Kang Li, Dianli Zhao, and Ceng Gao from the CryoEM facility for Marine Biology at QNLM for our cryo-EM data collection. We also thank the Electron Microscopy Facility of ShanghaiTech University for sample preparation and data collection. This work was supported by National Key R&D program of China, 2017YFA0504800 (to Q.S.), National Key R&D program of China, 2018YFC1406700 (to Q.S.), and National Natural Science Foundation of China, 31870743 (to Q.S.)

## Author contributions
N.Z. and Y.Q. expressed and purified proteins and prepared and screened EM grids. X.S., Y.Q., and N.Z. collected cryo-EM datasets. X.S. and H.S. did data processing and carried out model building and refinement with assistance from Y.L. L.M. and Y.B. did RNA gel assay. T.L., L.Y., and L.Q. helped with input and discussion during the course of the work. H.S., X.S., N.Z., Y.Z., T.L., and Q.-T.S. analyzed all the data. Q.-T.S. supervised the work. Q.-T.S. wrote the manuscript with input from N.Z., Y.Q., T.L., X.S., and H.S.

## Competing interests
The authors declare no competing interests.
