## [Peer Review File · Communications Biology]

Reviewers' comments:

Reviewer #2 (Remarks to the Author):

The manuscript Structural plasticity of mumps virus nucleocapsids for genome protection and condensation by Qing-Tao Shen et al describes reconstructions of MuV NC protein at near atomic resolution by cryo-EM in different ring and helical forms. The structures themselves are all at good resolution and most of the structural descriptions are therefore solid.

However, the manuscript is rather hard to read as the English and scientific writing is poor. Some serious editing is warranted. There are even points where the wrong structural form is described in the text e.g. the legend for Figure 4e says that: "Genome condensation from NChelix-dense to NCring- stacked." (line 430) And then "The heights of 390-nt RNA in NChelix-dense and NChelix-hyper are marked." The ring stacked structure is not in 4e. They also claim in the Abstract to report 3 ring structures and 3 helical structures (line 24) though they only report 2 ring structures. There are several of these kinds of mistakes which leave the reader rather confused to say the least. The figures and legends are generally rather unclear. Whilst I appreciate that it can be difficult to describe structures in text and to show relevant details, this could be significantly improved here. As an example, Figure 1 e and f are far too small and the structural interpretation that is supposed to be indicated is not clear at all. All figures need to be improved for clarity. Simple things such as having micrographs at the same scale for easy comparison would help (Supp Fig 5).

Having said that the structures and their descriptions are solid, I do however wonder how relevant they are. The dense helical form would appear to be the most "normal" form, and of course there must be transitions between encapsidated (helical) forms of RNPs and an unwound form that the polymerase will use during replication or transcription. As stated in the manuscript, the phosphoprotein can help in unwinding. Thus, the dense to hyperdense transition may be relevant, and the supplementary video is helpful here. However, these structures are all derived from protein expressed recombinantly in *E. coli*, and bring a short *E. coli* RNA segment with them through purification. RNPs are formed in an infected cell as the RNA is exiting the polymerase – indeed the Phosphoprotein can chaperone NC protein for this purpose. Therefore the likelihood of ring formation in the cell on short RNAs is very low, so it appears to me that rings are a result of the recombinant expression rather than anything "biological" i.e. what is happening in the infected cell. The same applies to the stacked rings, with the additional problem that there is no protected route from one flat circle of RNA within a stacked ring to the neighbouring ring. Exposure of naked genomic RNA between rings would seem like a poor strategy, unlikely to be adopted by the virus. All of this at the very least needs comment and discussion, but I fear that at least 3 of the 5 structures are not relevant for biological interpretation.

In addition, discussion of proteolytic digestion by contaminants in the prep from *E. coli* seems an over interpretation – unless it could be shown that the same is happening in an infected cell. I don't see how you can have a biological interpretation of degradation of the protein *in vitro* by non-relevant contaminants. The statement that "a new role was identified for the C-arm as the regulator of the transition between the dense and hyperdense states of MuV helical nucleocapsids." (line 201) is predicated on this over-interpretation.

"Such structural plasticity makes genome protection and genome condensation feasible, and provides potential targets for future therapies and better vaccination." (line 261) is another example. The potential as targets is a stretch and I have no idea how this might provide better vaccination.

"the long and rigid nucleocapsids might be a hibernation-like state in the stringent conditions remaining in host cells by tuning the genome protection" (line 276) is also rather wild speculation.

I cannot therefore recommend publication in the current form.

Reviewer #3 (Remarks to the Author):

Overview

Mumps virus is a negative strand RNA virus (NSV) belonging to the Paramyxoviridae family and the order mononegavirales. Structures of authentic mumps virus particles have been characterized before at lower resolutions and the intrinsic flexibility observed in the nucleocapsids prevent from obtaining high-resolution structures. The manuscript by Shen et al, describes the high-resolution structures of Nucleoprotein (N) in two different oligomeric states and four different higher-order helical structures. Three of the helical structures were obtained by prolonged incubation at 4C and their formation was due to the limited proteolysis of the C-terminal residues. The manuscript further describes the structural rearrangements required to transition between the different helical assemblies obtained. The high-resolution structures of the Nucleoprotein (N) in various states presented in the manuscript are a valuable addition to the field and could improve the understanding of NSV nucleocapsid assembly. However, the manuscript missed the opportunity to compare their structures to other known structures and make a convincing connection to the overall mumps virus nucleocapsid assembly. The impact of the manuscript on the field is unclear due to the lack of analysis and the impact looks incremental in the current form. Following are some of the major and minor issues with the manuscript.

Major issues

1. The manuscript is very limited in describing and citing work by other groups. The following manuscripts provide valuable insights into Paramyxovirus and NSV viral assemblies, which are relevant to the manuscript but not utilized.
 - <https://www.ncbi.nlm.nih.gov/pmc/articles/PMC250641/> - Different helical states of Sendai virus
 - <https://pubmed.ncbi.nlm.nih.gov/19726519/> - Effect of mutations of C-terminal residues of VSV nucleocapsid protein on viral assembly
 - <https://pubmed.ncbi.nlm.nih.gov/23677789/> - Structure of Respiratory syncytial virus nucleocapsid
 - <https://pubmed.ncbi.nlm.nih.gov/15201055/> - Conformational flexibility of Measles virus nucleocapsid
 - <https://pubmed.ncbi.nlm.nih.gov/17459940/> - Role of C-terminal domain of Sendai virus nucleocapsid in interaction with Phosphoprotein P.
 - <https://pubmed.ncbi.nlm.nih.gov/18003727/> - VSV nucleoprotein structure solved in an oligomeric state and analysis of different mutants.
 - <https://pubmed.ncbi.nlm.nih.gov/25108352/> - Nipah virus nucleoprotein structure in complex with the chaperone.
 - <https://pubmed.ncbi.nlm.nih.gov/24429372/> - Review comparing different Negative strand virus nucleocapsids
2. The helical reconstruction requires an initial estimate of the twist and rise of the helix. Usually, diffraction pattern of 2D class average images are used for determining the initial twist and rise, and later compared with the actual reconstruction to make sure that the reconstruction was not a local minima. The manuscript should show the diffraction pattern of the 2D class averages and compare them to the obtained 3D reconstructions, for all the helical structures in supplemental data. This will also improve the reproducibility of the data.
3. Lack of comparison to other known structures of nucleocapsid proteins like VSV, etc is disappointing. There are also structures of mumps NLP and authentic mumps viral particles at lower resolutions and the manuscript does not compare their structures to them. This will help in highlighting the contribution of this manuscript to the field and help in improving the general understanding of virus assembly. Will suggest looking into the review by (Green et al, J. Virol, 2014 Apr;88(7):3766-75) for an overview.
4. From a previous study on authentic mumps virus particles it was shown that the phosphoprotein P can unwind the helical nucleocapsid by interacting with the nucleocapsid protein N. Will the helical structures form in the presence of P? Do the different helical structures obtained in the manuscript still be susceptible to unwinding by P protein as they lack the C-terminal residues?
5. How important are the residues (375 -549) of N for nucleocapsid assembly in-vivo? Which can highlight the significance of this region by in viral replication. This can be addressed by comparing

the ability of WT N and mutant N to support viral replication utilizing the minigenome system described in the previous mumps virus work (Cox et al, PNAS, 2014 Oct 21;111(42):15208-13). 6. One of the major questions in the field is how the nucleocapsid protein is prevented from oligomerizing into ring like structures like the ones described in the manuscript. In the discussion (Pg 9, line 11,12,13) the manuscript states that the detailed mechanism of transition from ring like structures to helical nucleocapsids is missing. But the N protein never exists as a ring like structure in-vivo and rather is sequestered in a monomeric form by the P protein. Not clear what the manuscript is referring to in this statement.

Minor issues

1. The use of term nucleocapsid to describe the different structures obtained is misleading and confusing. The manuscript should use the convention of denoting the nucleocapsid protein as N in the free form and NOT NC, similar to what they did in their Newcastle disease virus work. Further, it might be acceptable to refer the helical structures as NC if they are clearly described as nucleocapsid like particles somewhere in the manuscript.
2. The EMDB deposition does not include the unfiltered half maps, masks used for RELION post-processing and the B-factor used for sharpening the maps. It's a good practice in the field and improves the overall quality of the data obtained.
3. The manuscript stated that the C-arm is invisible in their reconstructions of the ring structures and goes on to speculate that it might be flexible. The manuscript does not specifically describe in which members of the mononegavirales order the C-arm is resolved and what role it plays.

Reviewer #4 (Remarks to the Author):

This manuscript describes a structural analysis of the mumps virus (MuV) nucleocapsid-like particles, produced by heterologous expression.

The paramyxovirus nucleocapsid (NC) is a helical ribonucleoprotein assembly comprising the viral genome and a viral encoded protein N. Many structural studies have been performed over the past twenty years, exploiting heterologous expression of N, whereby non-specific encapsidation of cellular RNA leads to formation of both helices and rings. High-resolution structure analysis of helices has been hampered by flexibility of helices comprising full-length N protein, consequently many investigators have exploited the fact that the paramyxovirus NC becomes rigid upon proteolytic cleavage of the C-terminus. In this way both rings and rigid helix structures have been solved at near-atomic resolution.

Here the authors exploit both approaches to produce a series of high-resolution structures in which novel features are identified – in particular loss of a structured C-terminal arm and rearrangement of the N-terminal arm is noted in both rings and densely coiled helices. These different structures are hypothesised to have biological relevance in the assembly of the NC.

While these data are interesting and the resolution achieved for the C-terminally truncated protein is impressive, I question whether it is reasonable to infer biological relevance from these structures as they reflect forms of the NC that would not be expected to exist in authentic virus infection.

The formation of rings upon heterologous expression is very much more common in bacterial expression systems, while expression in insect cells more frequently leads to the production of long-helical NC-like particles. This is possibly a consequence of the nature of encapsidated cellular RNA. In this study the encapsidation of 80nt RNA molecules does not reflect the situation in viral replication, whereby N proteins, chaperoned by P, are sequentially added to the nascent RNA genome, which is eventually ~15 kb long. Thus, a ring should never exist in vivo (contrary to the statement in paragraph 15 page 5). We should not then expect stacks of rings to exist either. Likewise, the formation of dense rigid NCs as a consequence of proteolytic degradation of the C-terminus of N is also not a biologically relevant structure. The C-terminal arm is a critical component of N, mediating the interaction with the phosphoprotein and thereby the RNA-

dependent RNA polymerase. Removing it leads to a structure that can never exist in vivo. While it is the case that the interaction between P and the Ntail modulates the NC structure – the changes range from a helix of pitch ~ 56 angstroms to a loosely coiled structure.

Many of the findings and approaches described in this article were first described and pioneered by my lab, although we are not cited. While I hate to be the reviewer that requests citations for my own work, in this instance I believe that it is justifiable, and I shall therefore do so openly by signing this review:

Description of 9-14 subunit rings and descriptions of the range of helical pitch in paramyxovirus nucleocapsids were the subject of our paper: Significant differences in nucleocapsid morphology within the Paramyxoviridae D Bhella, A Ralph, LB Murphy, RP Yeo - *Journal of General Virology*, 2002, indeed the variation of helical pitch in nucleocapsids was noted far earlier for example in the study from Ed Egelman of the Sendai virus NC structure (The Sendai virus nucleocapsid exists in at least four different helical states. *Journal of Virology* 1989), and from Heggeness et al *PNAS* 1980. We also described variation in helical pitch and twist along a single paramyxovirus nucleocapsid (by tracing segments from real-space helical reconstruction back to the micrograph) in our paper: Conformational flexibility in recombinant measles virus nucleocapsids visualised by cryo-negative stain electron microscopy and real-space helical reconstruction, D Bhella, A Ralph, RP Yeo - *Journal of molecular biology*, 2004. This latter study directly addressed (and contradicts) some of the points made in paragraph 20 of page 4.

In summary, while it is manifestly true that heterologous expression of N and subsequent removal of the C-terminal sequences involved in P binding (Ntail), have allowed for high-resolution analysis of the Ncore structure, extreme care should be taken not to over interpret the conformations of the resulting assemblies.

Reviewed by David Bhella

Reviewer #2 (Remarks to the Author):

The manuscript Structural plasticity of mumps virus nucleocapsids for genome protection and condensation by Qing-Tao Shen et al describes reconstructions of MuV NC protein at near atomic resolution by cryo-EM in different ring and helical forms. The structures themselves are all at good resolution and most of the structural descriptions are therefore solid.

However, the manuscript is rather hard to read as the English and scientific writing is poor. Some serious editing is warranted. There are even points where the wrong structural form is described in the text e.g. the legend for Figure 4e says that: “Genome condensation from NChelix-dense to NCring- stacked.” (line 430) And then “The heights of 390-nt RNA in NChelix-dense and NChelix-hyper are marked.” The ring stacked structure is not in 4e. They also claim in the Abstract to report 3 ring structures and 3 helical structures (line 24) though they only report 2 ring structures. There are several of these kinds of mistakes which leave the reader rather confused to say the least. The figures and legends are generally rather unclear. Whilst I appreciate that it can be difficult to describe structures in text and to show relevant details, this could be significantly improved here. As an example, Figure 1 e and f are far too small and the structural interpretation that is supposed to be indicated is not clear at all. All figures need to be improved for clarity. Simple things such as having micrographs at the same scale for easy comparison would help (Supp Fig 5).

Re:

Thanks for reviewing our manuscript.

Thanks for pointing out the mistake on “N_{Cring-stacked}” in Figure 4. We have corrected “N_{Cring-stacked}” to “N_{Chelix-hyper}”. We also followed the advice and have polished other confusing descriptions in the manuscript.

As to the number of different MuV nucleoprotein assemblies, we totally resolved 6 high-resolution structures: N_{ring-13p}, N_{ring-14p}, N_{ring-stacked}, N_{Chelix-dense}, N_{Chelix-hyper} and N_{Chelix-Δarm} (Supplementary Table 1). Even though N_{ring-stacked} stays as a filament, it consists of layers of rings and is thought not biologically relevant. Based on this, we classified and named N_{ring-stacked} as a kind of ring structure, and the number of ring-like structures was counted at 3 instead of 2 in our initial manuscript. Just as what the reviewer mentioned, our previous description in the “Abstract” was confusing. So, we directly listed the numbers of each MuV nucleoprotein assemblies in the revised manuscript (see line 28-30).

Thanks for great suggestions on figure preparation. We have revised all figures and figure legends including Figure 1 and Supplementary Fig. 5. Thanks.

Having said that the structures and their descriptions are solid, I do however wonder how relevant they are. The dense helical form would appear to be the most “normal” form, and of

course there must be transitions between encapsidated (helical) forms of RNPs and an unwound form that the polymerase will use during replication or transcription. As stated in the manuscript, the phosphoprotein can help in unwinding. Thus, the dense to hyperdense transition may be relevant, and the supplementary video is helpful here. However, these structures are all derived from protein expressed recombinantly in *E. coli*, and bring a short *E. coli* RNA segment with them through purification. RNPs are formed in an infected cell as the RNA is exiting the polymerase – indeed the Phosphoprotein can chaperone NC protein for this purpose. Therefore the likelihood of ring formation in the cell on short RNAs is very low, so it appears to me that rings are a result of the recombinant expression rather than anything “biological” i.e. what is happening in the infected cell. The same applies to the stacked rings, with the additional problem that there is no protected route from one flat circle of RNA within a stacked ring to the neighbouring ring. Exposure of naked genomic RNA between rings would seem like a poor strategy, unlikely to be adopted by the virus. All of this at the very least needs comment and discussion, but I fear that at least 3 of the 5 structures are not relevant for biological interpretation.

Re:

Thanks for the great comments.

The improper description about “transition from N-RNA rings to helical nucleocapsids”: *In vivo*, the P protein will sequester the N protein in a monomeric N_0P form, N_0P instead of N-RNA rings will be utilized as the building blocks for the nucleocapsid assembly. The recombinant MuV nucleoproteins assemble into N-RNA rings, and the addition of trypsin into MuV N-RNA rings will yield helical nucleocapsids. Our description about “the transition from N-RNA rings to helical nucleocapsids” was confusing and even misleading. We have taken the advice and corrected the improper statements in the revised manuscript (see *line 30-34, line 152-156*).

Biologically irrelevant MuV N-RNA rings: We totally agree with the reviewer that MuV N-RNA rings ($N_{\text{ring-13p}}$ and $N_{\text{ring-14p}}$) and $N_{\text{ring-stacked}}$ should not be biologically relevant, because it would be impossible for genomic RNA to cross stacked rings (see *line 142-143*). Helical nucleocapsids such as $NC_{\text{helix-dense}}$ and $NC_{\text{helix-hyper}}$ offer the opportunity to enwrap the embedded RNA for genome encapsulation and protection, which are supposed closer-to-native. In the revised manuscript, we follow the reviewer’s suggestion and use N-RNA rings ($N_{\text{ring-13p}}$, $N_{\text{ring-14p}}$ and $N_{\text{ring-stacked}}$) and helical nucleocapsids ($NC_{\text{helix-dense}}$ and $NC_{\text{helix-hyper}}$) to distinguish biological relevance or not.

Structural plasticity deduced from both MuV N-RNA rings and helical nucleocapsids:

Similar to the previous report (Cox *et al.*, *Journal of Virology* 2009), purified full-length MuV nucleoproteins from *E. coli* stay in ring-like structures. Other nucleoproteins from RSV, VSV and the Rabies virus *et al.* are also purified and resolved as rings under X-ray crystallography. How to prevent the oligomerization of nucleoproteins into rings is a major question in the field. Biochemical and structural analysis indicates that C-terminal tail (N-tail) of the MuV nucleoprotein hampers the assembly of helical nucleocapsids. The loss of N-tail from MuV nucleoprotein will change from rings to dense nucleocapsids ($NC_{\text{helix-dense}}$) with the helical pitch at 5.6 nm. The further removal of C-arm will transit dense MuV

nucleocapsid (NC_{helix-dense}) into hyperdense nucleocapsid (NC_{helix-hyper}) with the helical pitch at 4.5 nm. Thus, N-tail hampers the assembly of helical nucleocapsids, and C-arm directly regulates the transition between the dense and hyperdense states of helical nucleocapsids. So, the *in vitro* high-resolution structures on both ring-like structures and helical nucleocapsids are meaningful and help a lot to clarify the molecular mechanism of structural plasticity.

Regulation of structural plasticity via P proteins or others: Just as the reviewer mentioned, the assembly of helical nucleocapsids *in vivo* is tightly related to other viral or host proteins especially P protein, and the switch between loose, dense and hyperdense nucleocapsids might also be regulated by viral or host proteins such as P protein. The detailed mechanism is worth thorough investigation via resolving the complex of P protein and nucleoprotein via cryo-EM.

Thanks.

In addition, discussion of proteolytic digestion by contaminants in the prep from E. coli seems an over interpretation – unless it could be shown that the same is happening in an infected cell. I don't see how you can have a biological interpretation of degradation of the protein in vitro by non-relevant contaminants. The statement that “a new role was identified for the C-arm as the regulator of the transition between the dense and hyperdense states of MuV helical nucleocapsids.” (line 201) is predicated on this over-interpretation.

Re:

Thanks for the criticism.

We admitted that we over-interpreted the role of C-arm in the regulation of the transition between dense and hyperdense states of MuV helical nucleocapsids. We have deleted the improper statements (see *line 269-303*) from the “Discussion”.

Thanks.

“Such structural plasticity makes genome protection and genome condensation feasible, and provides potential targets for future therapies and better vaccination.” (line 261) is another example. The potential as targets is a stretch and I have no idea how this might provide better vaccination.

Re:

Thanks for the criticism.

In influenza virus, nucleoprotein has been proposed as a good target for a universal vaccine (Christopher *et al*, *Antiviral Chemistry and Chemotherapy* 2013). Inhibitors targeting the nucleoprotein have also been identified for respiratory syncytial virus. For instance, RSV604 is a benzodiazepine derivative developed from a hit identified in an antiviral cell protection assay (Henderson *et al*. *Journal of Medical Chemistry*, 2007; Chapman *et al*, *Antimicrob Agents Chemother* 2007). Inspired by these results, we intend to make full use of structural information from high-resolution MuV nucleoproteins and specifically design epitopes for vaccination and therapy. We took the reviewer's criticism and have removed the over-interpreted statement in our revised manuscript.

Thanks.

“the long and rigid nucleocapsids might be a hibernation-like state in the stringent conditions remaining in host cells by tuning the genome protection” (line 276) is also rather wild speculation.

Re:

Thanks for the criticism.

We have followed the reviewer’s advice and removed the statement from our revised manuscript. Thanks.

Reviewer #3 (Remarks to the Author):

Overview

Mumps virus is a negative strand RNA virus (NSV) belonging to the Paramyxoviridae family and the order mononegavirales. Structures of authentic mumps virus particles have been characterized before at lower resolutions and the intrinsic flexibility observed in the nucleocapsids prevent from obtaining high-resolution structures. The manuscript by Shen et al, describes the high-resolution structures of Nucleoprotein (N) in two different oligomeric states and four different higher-order helical structures. Three of the helical structures were obtained by prolonged incubation at 4C and their formation was due to the limited proteolysis of the C-terminal residues. The manuscript further describes the structural rearrangements required to transition between the different helical assemblies obtained. The high-resolution structures of the Nucleoprotein (N) in various states presented in the manuscript are a valuable addition to the field and could improve the understanding of NSV nucleocapsid assembly. However, the manuscript missed the opportunity to compare their structures to other known structures and make a convincing connection to the overall mumps virus nucleocapsid assembly. The impact of the manuscript on the field is unclear due to the lack of analysis and the impact looks incremental in the current form. Following are some of the major and minor issues with the manuscript.

Major issues

1. The manuscript is very limited in describing and citing work by other groups. The following manuscripts provide valuable insights into Paramyxovirus and NSV viral assemblies, which are relevant to the manuscript but not utilized.

- <https://www.ncbi.nlm.nih.gov/pmc/articles/PMC250641/> - Different helical states of Sendai virus*
- <https://pubmed.ncbi.nlm.nih.gov/19726519/> - Effect of mutations of C-terminal residues of VSV nucleocapsid protein on viral assembly*
- <https://pubmed.ncbi.nlm.nih.gov/23677789/> - Structure of Respiratory syncytial virus nucleocapsid*
- <https://pubmed.ncbi.nlm.nih.gov/15201055/> - Conformational flexibility of Measles virus nucleocapsid*
- <https://pubmed.ncbi.nlm.nih.gov/17459940/> - Role of C-terminal domain of Sendai virus nucleocapsid in interaction with Phosphoprotein P.*
- <https://pubmed.ncbi.nlm.nih.gov/18003727/> - VSV nucleoprotein structure solved in an oligomeric state and analysis of different mutants.*
- <https://pubmed.ncbi.nlm.nih.gov/25108352/> - Nipah virus nucleoprotein structure in complex with the chaperone.*
- <https://pubmed.ncbi.nlm.nih.gov/24429372/> - Review comparing different Negative strand virus nucleocapsids*

Re:

Thanks for reviewing our manuscript. And also, thanks so much for listing the related literatures.

We have followed the great advice and compared our structures with other known structures from Measles virus, VSV, PIV5, Nipah virus and Sendai virus *et al* in the “Discussion” (see *line 269-290*). Just as what the reviewer expected, the structural comparison provides valuable insights into Paramyxovirus and NSV viral assemblies, which raises the impact of our work in the field.

2. The helical reconstruction requires an initial estimate of the twist and rise of the helix. Usually, diffraction pattern of 2D class average images are used for determining the initial twist and rise, and later compared with the actual reconstruction to make sure that the reconstruction was not a local minima. The manuscript should show the diffraction pattern of the 2D class averages and compare them to the obtained 3D reconstructions, for all the helical structures in supplemental data. This will also improve the reproducibility of the data.

Re:

Thanks for the great suggestion.

Diffraction patterns have been analyzed on selected 2D class averages from $N_{\text{ring-stacked}}$, $NC_{\text{helix-dense}}$ and $NC_{\text{helix-hyper}}$. The rises of the helix derived from the diffraction analysis fit well to the values from the respective 3D reconstructions, and these results are listed in the P2P_Figure 1 (below) and Supplementary Figure 6a.

P2P_Figure 1. 2D class average images of MuV $NC_{\text{helix-dense}}$, $NC_{\text{helix-hyper}}$ and $N_{\text{ring-stacked}}$ and their diffraction patterns

*3. Lack of comparison to other known structures of nucleocapsid proteins like VSV, etc is disappointing. There are also structures of mumps NLP and authentic mumps viral particles at lower resolutions and the manuscript does not to compare their structures to them. This will help in highlighting the contribution of this manuscript to the field and help in improving the general understanding of virus assembly. Will suggest looking into the review by (Green *et al*, *J. Virol*, 2014 Apr;88(7):3766-75) for an overview.*

Re:

We really appreciate the wonderful comments from the reviewer.

Just as what the reviewer mentioned, the previous structures of mumps virus nucleocapsids especially from authentic mumps viral particles are landmarks in mumps virus field. We have followed the advice and emphasized the previously published structures for comparison (see *line 76-84*), which indeed helps us improve the understanding of mumps virus assembly.

Previous studies show that recombinant VSV nucleoproteins also assemble into a closed ring with 10 protomers under EM (*Chen et al., Structure 2004*). Immediately following this work, Prof. Ming Luo's group reported the high-resolution ring-like structure on VSV nucleoprotein via X-ray crystallography. In the crystallographic unit cell, there are two VSV N-RNA rings, which are packed in a head-to-head manner. Different from VSV, MuV N_{ring-stacked} comprises of layers of rings packed in a head-to-tail mode. This difference is mentioned in the revised manuscript (see *line 163-166*). Thanks for the great suggestion.

4. From a previous study on authentic mumps virus particles it was shown that the phosphoprotein P can unwind the helical nucleocapsid by interacting with the nucleocapsid protein N. Will the helical structures form in the presence of P? Do the different helical structures obtained in the manuscript still be susceptible to unwinding by P protein as they lack the C-terminal residues?

Re:

Thanks for the great comments.

P protein is reported to have multiple functions such as binding to N₀ to prevent self-assembly of N₀, unwinding helical nucleocapsid, and facilitating the transcription and replication of L. Previous studies show that MuV P_{NTD} can induce full-length MuV helical nucleocapsid to uncoil into a large of number unwound nucleocapsid segments (*Cox et al., PNAS 2014*). These unwound nucleocapsids are easily accessed by L during genome replication and transcription.

Considering that P protein usually recognizes the N-tail of nucleoproteins and then recruit L to initiate genome transcription and replication. In our MuV N_{helix-dense} and N_{helix-hyper}, N-tails are cleaved off by residual impurities to reduce the structural flexibility (Fig. 2a). Thus, our speculation is that P protein might not bind or unwind N_{helix-dense} and N_{helix-hyper}. Actually, full-length MuV nucleoproteins keep their N-tails and prefer assembling into ring-like structures. It will be interesting to check whether P protein can switch N-RNA rings to helical nucleocapsids or whether P protein and the subsequent L protein can bind N-RNA rings to assemble into N-P-L supercomplex. Following this idea, we have co-expressed MuV P-L in insect cells and purified P-L complex via tandem affinity and size-exclusion chromatography (P2P_Figure 2). We will incubate P-L complex with MuV N-RNA rings. The samples will be subjected to cryo-EM for high-resolution reconstruction. Thanks.

P2P_Figure 2. Purification of MuV P-L complex from insect cells

5. *How important are the residues (375 -549) of N for nucleocapsid assembly in-vivo? Which can highlight the significance of this region by in viral replication. This can be addressed by comparing the ability of WT N and mutant N to support viral replication utilizing the minigenome system described in the previous mumps virus work (Cox et al, PNAS, 2014 Oct 21;111(42):15208-13).*

Re:

Thanks for great suggestion.

Residues from 375 to 549 of MuV N consist of C-arm and N-tail domains. Previous work shows that C-arm, together with N-arm, plays roles in the assembly of helical nucleocapsids from many mononegaviruses. Comparing to N-arm, C-arm seems not a requisite in the oligomerization of nucleoproteins, because either C-arm or N-tail is not visible in cryo-EM maps in MuV N-RNA rings and NC_{helix-hyper}, and the truncation of 375-549 on MuV N can still assemble into helical structures (NC_{helix-Δarm}) (see *line 203-210*). The relocation of C-arm by P protein or others might not affect the assembly of MuV helical nucleocapsid but influence the viral genome replication and transcription.

N-tail is also important to regulate genome transcription and replication via interacting with P protein. Our minigenome assay on NDV shows that N-tail deletion mutation will dramatically influence viral replication, and deletion of both C-arm and N-tail (residues from 370 to 489) will totally abolish the replication (*Song et al. eLife 2019*). Considering the structural similarity of nucleocapsids between NDV and MuV, we believe that residues from 375 to 549 in MuV nucleoprotein will influence the mumps viral replication.

Thanks.

6. *One of the major questions in the field is how the nucleocapsid protein is prevented from oligomerizing into ring like structures like the ones described in the manuscript. In the discussion (Pg 9, line 11,12,13) the manuscript states that the detailed mechanism of transition from ring like structures to helical nucleocapsids is missing. But the N protein never exists as a ring like structure in-vivo and rather is sequestered in a monomeric form by the P protein. Not clear what the manuscript is referring to in this statement.*

Re:

Thanks for great suggestion.

Just as what the reviewer mentioned that the P protein will sequester the N protein in a monomeric N₀P form, N₀P instead of N-RNA rings will be utilized as the building blocks for the nucleocapsid assembly. Our description about “the transition from N-RNA rings to helical nucleocapsids” was confusing and even misleading. We have taken the advice and corrected the improper statements in the revised manuscript (see *line 30-34, line 274-276*).

Similar to previous reports, the recombinant MuV nucleoproteins assemble into N-RNA rings *in vitro*. Interestingly, the addition of trypsin into N-RNA rings will yield helical nucleocapsids. Our high-resolution structures on MuV N-RNA rings and helical nucleocapsids indicate that N-tail, as a negative regulator, might hamper the assembly of helical nucleocapsids. *In vivo*, other viral or host proteins such as P protein might relocate N-tail to assemble into helical nucleocapsids. Just as what we mentioned before, we are checking whether P-L complex can remodel N-RNA rings into helical nucleocapsids (P2P_Figure 2). Thanks.

Minor issues

1. The use of term nucleocapsid to describe the different structures obtained is misleading and confusing. The manuscript should use the convention of denoting the nucleocapsid protein as N in the free form and NOT NC, similar to what they did in their Newcastle disease virus work. Further, it might be acceptable to refer the helical structures as NC if they are clearly described as nucleocapsid like particles somewhere in the manuscript.

Re:

Thanks for pointing out this issue.

We have followed the advice and only kept helical nucleocapsid-like particles as NC. All the other ring-like structures and stacked rings of nucleoprotein-RNA complexes are named as N_{ring-13p}, N_{ring-14p} and N_{ring-stacked}, respectively.

2. The EMDB deposition does not include the unfiltered half maps, masks used for RELION post-processing and the B-factor used for sharpening the maps. It's a good practice in the field and improves the overall quality of the data obtained.

Re:

Thanks for the suggestions.

We have deposited our unfiltered half maps, masks for RELION post-processing and B-factor to the EMDB.

3. The manuscript stated that the C-arm is invisible in their reconstructions of the ring structures and goes on to speculate that it might be flexible. The manuscript does not specifically describe in which members of the mononegavirales order the C-arm is resolved and what role it plays.

Re:

Thanks for the comments.

N-arm and C-arm plays an important role in holding neighboring protomers together via a domain swapping mode, as revealed in many mononegaviruses such as the Rabies virus, VSV, PIV5, MeV and RSV. Interestingly, C-arm seems not as essential as N-arm in the assembly of nucleoprotein-RNA complex, which are reported in PIV5 and measles virus (Aggarwal *et al. Journal of Virology*, 2018; Guryanov *et al. Journal of Virology*, 2015). The missing of C-arm in the EM density of full-length MuV N-RNA rings is speculated flexible and not involved in the interaction between neighboring protomers.

We have listed members of the mononegavirales in the revised manuscript, whose C-arms are resolved via either X-ray crystallography or cryo-EM. We also pointed out the role of C-arm in the assembly of N-RNA more clearly (see *line 98-107*). Thanks.

Reviewer #4 (Remarks to the Author):

This manuscript describes a structural analysis of the mumps virus (MuV) nucleocapsid-like particles, produced by heterologous expression.

The paramyxovirus nucleocapsid (NC) is a helical ribonucleoprotein assembly comprising the viral genome and a viral encoded protein N. Many structural studies have been performed over the past twenty years, exploiting heterologous expression of N, whereby non-specific encapsidation of cellular RNA leads to formation of both helices and rings. High-resolution structure analysis of helices has been hampered by flexibility of helices comprising full-length N protein, consequently many investigators have exploited the fact that the paramyxovirus NC becomes rigid upon proteolytic cleavage of the C-terminus. In this way both rings and rigid helix structures have been solved at near-atomic resolution.

Here the authors exploit both approaches to produce a series of high-resolution structures in which novel features are identified – in particular loss of a structured C-terminal arm and rearrangement of the N-terminal arm is noted in both rings and densely coiled helices. These different structures are hypothesised to have biological relevance in the assembly of the NC.

While these data are interesting and the resolution achieved for the C-terminally truncated protein is impressive, I question whether it is reasonable to infer biological relevance from these structures as they reflect forms of the NC that would not be expected to exist in authentic virus infection.

Re:

Thanks so much for reviewing our manuscript.

Just as what the reviewer mentioned that N proteins from the family of *Paramyxoviridae* will become straight for high-resolution structural analysis after proteolytic cleavage on flexible regions especially C-terminal tail (N-tail). Following this approach, we have resolved 6 high-resolution structures: N_{ring-13p}, N_{ring-14p}, N_{ring-stacked}, NC_{helix-dense}, NC_{helix-hyper} and NC_{helix-Δarm} via cryo-EM. Our series of high-resolution structures offer opportunity to study the molecular mechanism for the structural plasticity among MuV ring-like structures, dense nucleocapsid and hyperdense nucleocapsid, and this kind of structural plasticity is also shared by other paramyxoviruses.

Even though we have no cryo-ET data on the intact MuV virion due to the strictly controlled lab access, the structural plasticity in MuV N-RNA is speculated existing in authentic virus infection as in measles virus (MeV). Previous cryo-ET analysis on nucleocapsids from MeV virion shows a helical pitch at ~6.4 nm, in line with the values (5.0-6.6 nm) from recombinant nucleocapsids (*Liljeroos et al, PNAS 2011; Bhella et al, Journal of Molecular Biology, 2004*). This hints us that the recombinant nucleocapsids including MuV NC_{helix-dense} and NC_{helix-hyper} are probably biologically relevant, and structural plasticity is inherent in MuV nucleocapsids, which can regulate genome transcription and replication in the presence of viral or host proteins.

Certainly, it is very important and urgent for us to check whether the *in vitro* structural plasticity own by MuV, MeV, SeV and other viruses will occur in the authentic virus infection. Considering that SeV virion is easily accessed and operated in routine cryo-EM labs, we are using cryo-ET to shoot high-resolution nucleocapsids from SeV virion. (P2P_Figure 3)

P2P_Figure 3. Typical cryo-EM micrographs of intact SeV virion with fiducial gold particles

The formation of rings upon heterologous expression is very much more common in bacterial expression systems, while expression in insect cells more frequently leads to the production of long-helical NC-like particles. This is possibly a consequence of the nature of encapsidated cellular RNA. In this study the encapsidation of 80nt RNA molecules does not reflect the situation in viral replication, whereby N proteins, chaperoned by P, are sequentially added to the nascent RNA genome, which is eventually ~15 kb long. Thus, a ring should never exist in vivo (contrary to the statement in paragraph 15 page 5). We should not then expect stacks of rings to exist either. Likewise, the formation of dense rigid NCs as a consequence of proteolytic degradation of the C-terminus of N is also not a biologically relevant structure. The C-terminal arm is a critical component of N, mediating the interaction with the phosphoprotein and thereby the RNA-dependent RNA polymerase. Removing it leads to a structure that can never exist in vivo. While it is the case that the interaction between P and the Ntail modulates the NC structure – the changes range from a helix of pitch ~56 angstroms to a loosely coiled structure.

Re:

Thanks for the great comments.

The improper description about “transition from N-RNA rings to helical nucleocapsids”: We totally agree with the reviewer that MuV N-RNA rings (N_{ring-13p} and N_{ring-14p}) are not biologically relevant and the transition from ring-like structures to helical nucleocapsid will not occur in authentic virus infection. Even though N_{ring-stacked} stays as a filament, N_{ring-stacked} is composed of stacked rings, packing in a head-to-tail mode. Based on this, we classified and named N_{ring-stacked} as a ring-like structure, which is not biologically relevant, either.

In vivo, the P protein will sequester the N protein in a monomeric N_{0P} form, N_{0P} instead of N-RNA rings will be utilized as the building blocks for the nucleocapsid assembly. The

recombinant MuV nucleoproteins assemble into N-RNA rings, and the addition of trypsin into MuV N-RNA rings will yield helical nucleocapsids. Our description about “the transition from N-RNA rings to helical nucleocapsids” was confusing. We have taken the advice and corrected the improper statements in the revised manuscript (see *line 30-34, line 274-276*).

Relocation of N-tail/C-arm by P protein or others: For many mononegaviruses, N-tail of nucleoproteins has intrinsically disordered regions and brings great structural flexibility to helical nucleocapsids. Proteolytic cleavage on flexible N-tail seems the most popular manner to shoot high-resolution structures of nucleoproteins. Just as what the reviewer mentioned, the interaction between the P protein and N-tail is a biologically relevant manner to modulate the nucleocapsid structure with the changes from a helix of pitch at ~5.6 nm to a loosely coiled structure.

It is more interesting that the removal of both N-tail and C-arm will yield a hyperdense MuV nucleocapsids. C-arm is usually crucial for nucleocapsid assembly in VSV, RSV, PIV5 and MeV *et al.*, together with N-terminal arm, via a domain swapping manner. To our surprise, C-arms in MuV N-RNA rings, NC_{helix-hyper} and NC_{helix-Δarm} are not involved in nucleoprotein oligomerization. This hints that the relocation of C-arm by viral or host proteins such as P protein might show effect only on helical pitches. The changes in helical pitches are speculated to affect genome transcription/replication co-opted by phosphoprotein and RNA-dependent RNA polymerase. The detailed mechanism is really worthy of further investigation.

Even though some of MuV N-RNA structures from recombinant system are not biologically relevant, the distinct conformations including rings, packed rings, dense nucleocapsid and hyperdense nucleocapsid offer us great opportunity to study the detailed molecular mechanism for structural plasticity, which is shared by many mononegaviruses.

Thanks.

Many of the findings and approaches described in this article were first described and pioneered by my lab, although we are not cited. While I hate to be the reviewer that requests citations for my own work, in this instance I believe that it is justifiable, and I shall therefore do so openly by signing this review:

Description of 9-14 subunit rings and descriptions of the range of helical pitch in paramyxovirus nucleocapsids were the subject of our paper: Significant differences in nucleocapsid morphology within the Paramyxoviridae D Bhella, A Ralph, LB Murphy, RP Yeo - Journal of General Virology, 2002, indeed the variation of helical pitch in nucleocapsids was noted far earlier for example in the study from Ed Egelman of the Sendai virus NC structure (The Sendai virus nucleocapsid exists in at least four different helical states. Journal of Virology 1989), and from Heggeness et al PNAS 1980. We also described variation in helical pitch and twist along a single paramyxovirus nucleocapsid (by tracing segments from real-space helical reconstruction back to the micrograph) in our paper: Conformational flexibility in recombinant measles virus nucleocapsids visualised by cryo-

negative stain electron microscopy and real-space helical reconstruction, D Bhella, A Ralph, RP Yeo - Journal of

molecular biology, 2004. This latter study directly addressed (and contradicts) some of the points made in paragraph 20 of page 4.

In summary, while it is manifestly true that heterologous expression of N and subsequent removal of the C-terminal sequences involved in P binding (Ntail), have allowed for high-resolution analysis of the Ncore structure, extreme care should be taken not to over interpret the conformations of the resulting assemblies.

Re:

Thank Professor Bhella for reviewing our manuscript.

We are so sorry that in our previous manuscript, we didn't cite the works originally from your lab and other labs. We have taken your advice, read the literature intensively and drawn our statements carefully based on the important works published before.

Thanks so much.

Reviewed by David Bhella

Reviewers' comments:

Reviewer #1 (Remarks to the Author):

The revised manuscript, text and figures, are a significant improvement on the previous draft. Many mistakes have been corrected and the authors now better describe the rings and recognise them as non-biologically relevant. However, I do not believe that the modified text reflects the concerns of all reviewers re biological relevance of the helical structures, and the comprehensive review and comparison against known structures from related viruses has not been presented. In this context it is difficult to support publication of the draft in its current form.

Reviewer #2 (Remarks to the Author):

The revised manuscript by Shan et al, has addressed several comments raised during the initial peer-review process. The revised manuscript now includes appropriate citations and discussion of work by other groups. Further, the manuscript addresses the fact that the ring like structures are not biologically relevant and rather focuses on the structural plasticity of the nucleoprotein assemblies. The three different helical assemblies solved in the current study may not represent a biologically relevant form but the resolution of the structures are impressive and the information gained from the structures are a valuable contribution to the field.

The following sentences need to be revised to improve clarity and avoid over interpretation.

Lines 146 – 150 : There are no ring like structures in the nucleocapsid assembly pathway.
Lines 295 – 297 : C-arm plays a crucial role in nucleocapsid assembly and the authors state the same in their rebuttal letter. Not clear what the authors are referring to when they state that C-arm is not essential.

Reviewer #3 (Remarks to the Author):

Having read the revised manuscript and rebuttal, many of my original concerns remain. Fundamentally the paper still proposes that conformational flexibility in MuV NCs is a function of the changes brought about by proteolytic removal of the C-terminal arm and N-tail, however this is not proven. The argument that reordering of these domains by P might be involved in structural changes in authentic viruses is at least plausible, but there are many reasons to suspect that the forms described are artifactual, not least the tight, rigid form of the helix has never been demonstrated in virus infected cells. The manuscript remains a very challenging read and many sections of text are opaque and difficult to understand.

Fundamentally of all the structures solved, only one NCdense, is potentially biologically relevant, inferring biological roles for observed plasticity is conjecture.

Some points that should be addressed:

Line 30 "Structural analysis indicates that C-terminal tail of MuV nucleoprotein hampers the assembly of helical nucleocapsids" clearly this is not the case, it does not hamper NC assembly in authentic virus infections.

Line 83 – dense and hyperdense states are not defined here or in the previous literature.

118 – text here is obscure – needs to be clear that forms discussed are a consequence of heterologous expression.

180 – Ring structures were solved using tilted data at 20 and 40 degrees – the map looks poor in

supplemental figure 3 – what was the angular distribution in these reconstructions, as there is still a large missing cone after tilting? Is there sufficient resolution in z? Is the resolution measured isotropic?

180 – SDS pages shows some degradation – are they sure that the C-arm is present?

188 - "Thus, the flexible C-arm of MuV Nring-13p and Nring-14p will not get involved in nucleoprotein oligomerization via forming a stable interface with the α 16 helix from the neighboring protomer" ... as has been shown for...(another virus?).

189-200- I found this text quite hard to follow and the legend of figure 1 was similarly challenging (and the figures very small).

1244. – How can the hyper-dense form help with packing – firstly it is rigid not flexible, secondly it can no longer function as a template without Ntail. The hyper-dense form solved cannot be a biologically relevant structure as it lacks critical regions of the N protein.

1251 – protection against nuclease digestions in the absence of C-terminal sequences of N is not biologically relevant - as the helix does not exist in this state in authentic viral infections (or at least has not been demonstrated to)

Supplemental figures do not appear to be revised or to show the analysis of layer lines in power spectra requested by reviewer 3

Reviewers' comments:

Reviewer #1 (Remarks to the Author):

The revised manuscript, text and figures, are a significant improvement on the previous draft. Many mistakes have been corrected and the authors now better describe the rings and recognise them as non-biologically relevant.

However, I do not believe that the modified text reflects the concerns of all reviewers re biological relevance of the helical structures, and the comprehensive review and comparison against known structures from related viruses has not been presented.

In this context it is difficult to support publication of the draft in its current form.

Re:

Thanks so much for reviewing our manuscript.

We have followed all reviewers' suggestions and revised our manuscript, accordingly.

In particular, we followed the reviewer's suggestion, introduced the structural studies on MuV nucleoproteins (mainly from Prof. Ming Luo's lab) in the *Introduction* (see line 62-72), compared our structures with authentic MuV nucleocapsids in the *Discussion* (see line 291-294), summarized the structural parameters of mononegaviral nucleoproteins (see line 294-295, P2P_Table 1, Supplementary Table 2) and reviewed the structural plasticity among MuV, SeV and MeV especially from Prof. Bhella's group (see line 295-302).

Thanks.

P2P_Table 1 Structural parameters of nucleocapsids in the order of *Mononegavirales*

Viruses	Nucleoproteins	Approach	Oligomeric States	Pitch (nm)	Twist	Resolution (Å)	EMDB PDB
MuV	Recombinant	cryo-EM	Ring	n/a	-27.7°	3.3	30282 7C30
				n/a	-25.7°	6.2	30281 7C30
			Helix	5.3	-27.1°	3.9	30264 7C1F
				4.6	-26.8°	3.6	30265 7C1F
	n/a	-27.7°	3.7	30266 7C1F			
Authentic	cryo-EM	Helix	6.7	-28.3°	18.1	2630 n/a	
NDV	Recombinant	cryo-EM	Clam	5.1	-27.5°	4.8	9793 6JC3
SeV	Recombinant	Negative stain EM	Helix	5.3 6.8 37.5	n/a	n/a	n/a n/a
		cryo-EM	Helix	5.4	-27.5°	4.1	30066 6M7D
				5.6	-27.4°	4.6	30065 6M7D
				5.3	-27.6°	2.9	30129 6M7D
			Clam	5.6	-27.1°	3.9	30064 6M7D

NiV	Recombinant	X-ray	C1	n/a	n/a	2.5	n/a 4CO6
		cryo-EM	Clam	n/a	-27.9°	4.3	n/a n/a
PIV5	Recombinant	X-ray	Ring	n/a	-27.7°	3.1	n/a 4XJN
RSV	Recombinant	X-ray	Ring	n/a	-36°	3.3	n/a 2WJ8
MeV	Recombinant	Cryo-negative stain EM	Helix	5 ~ 6.6	-26.8 ~ -27.6	n/a	n/a n/a
		cryo-EM	Helix	4.9	-29.2°	4.3	2867 4UFT
		X-ray	Ring	n/a	n/a	2.7	n/a 5E4V
	Authentic	cryo-ET	Helix	6.4	n/a	n/a	1973 n/a

Reviewer #2 (Remarks to the Author):

The revised manuscript by Shan et al, has addressed several comments raised during the initial peer-review process. The revised manuscript now includes appropriate citations and discussion of work by other groups. Further, the manuscript addresses the fact that the ring like structures are not biologically relevant and rather focuses on the structural plasticity of the nucleoprotein assemblies. The three different helical assemblies solved in the current study may not represent a biologically relevant form but the resolution of the structures are impressive and the information gained from the structures are a valuable contribution to the field.

Re:

Thanks so much for reviewing our manuscript.

With the great help from the reviewers, we have polished our manuscript and focused on structural plasticity of MuV N-RNA rings and helical nucleocapsids. Hope our work will benefit the field.

Thanks.

The following sentences need to be revised to improve clarity and avoid over interpretation.

Lines 146 – 150 : There are no ring like structures in the nucleocapsid assembly pathway.

Re:

Thanks so much for pointing out our error.

We agreed with the reviewer and removed the improper description about the transition from ring-like structures to helical nucleocapsids.

The revised sentence is that “*NTD or CTD of MuV phosphoproteins can directly interact with N-tails of MuV nucleoproteins and transform authentic MuV helical nucleocapsids into uncoiled or thicker helical nucleocapsids, respectively*” (see Line 154-156).

Thanks.

Lines 295 – 297 : C-arm plays a crucial role in nucleocapsid assembly and the authors state the same in their rebuttal letter. Not clear what the authors are referring to when they state that C-arm is not essential.

Re:

Sorry for the confusion on role of C-arm.

The domain swapping adopted by N-arm and C-arm is quite popular in the nucleocapsids assembly in many paramyxoviruses such as PIV5, MeV, SeV and NDV. Interestingly, C-arm seems not as essential as N-arm, because the removal of C-arm from NDV can still assemble into nucleocapsid filaments (Song *et al*, *eLife* 2019). Similarly, MuV NC_{helix-Δarm} does not have C-arm, but still assembles into helical nucleocapsids. Furthermore, N_{ring-13p} and N_{ring-14p} own C-arms, but C-arms are not resolved in high-resolution MuV N-RNA structures. Based on these, we made the statement that C-arm might not be as essential as N-arm. To avoid the confusion, we removed the description about the essential role of C-arm or not.

As what we mentioned in the manuscript, there is another unnoticed interface denoted as the swapped N-hole and N-loop, which is quite conserved in paramyxovirus nucleoproteins, which provides extra anchoring site for nucleocapsid assembly (see in P2P_Figure 1).

Thanks.

P2P_Figure 1. The swapped interface between the extended loops and N-holes in NDV, PIV5 and MeV.

Reviewer #3 (Remarks to the Author):

Having read the revised manuscript and rebuttal, many of my original concerns remain. Fundamentally the paper still proposes that conformational flexibility in MuV NCs is a function of the changes brought about by proteolytic removal of the C-terminal arm and N-tail, however this is not proven. The argument that reordering of these domains by P might be involved in structural changes in authentic viruses is at least plausible, but there are many reasons to suspect that the forms described are artifactual, not least the tight, rigid form of the helix has never been demonstrated in virus infected cells.

The manuscript remains a very challenging read and many sections of text are opaque and difficult to understand.

Fundamentally of all the structures solved, only one NCdense, is potentially biologically relevant, inferring biological roles for observed plasticity is conjecture.

Re:

Thanks so much for reviewing our manuscript.

We totally agree with the reviewer that MuV N-RNA rings and stacked rings are not biological relevant. The comparison between MuV N_{ring-13p} and N_{ring-14p} gives us hints that N-RNA rings exhibit structural plasticity. Furthermore, both MuV NC_{helix-dense} and NC_{helix-hyper} verify the occurrence of structural plasticity in helical nucleocapsids, which is very similar to what reported before in MeV nucleocapsids (Bhella et al. J. Mol. Bio. 2004). Our high-resolution structures for MuV N-RNA and helical nucleocapsids provide molecular mechanism for the in-plane and out-of-plane rotations from each protomer, which probably compose the basic movements for MuV structural plasticity.

Flexible N-tail of MuV nucleoproteins hampers structural determination on the full-length level. Trypsin treatment or residual impurities cleavage is a compromising approach to shoot high-resolution structures for almost all nucleoproteins. The high-resolution MuV NC_{helix-dense} and NC_{helix-hyper} after cleavage helps us point out the molecular mechanism for the structural plasticity. Just as what the reviewer mentioned that MuV nucleoprotein, together with phosphoprotein, will re-arrange the positions of its N-tail to provide structural flexibility. One of our on-going projects is to work on MuV N-P-L supercomplex (see in P2P_Figure 2).

We sincerely appreciate the intensive reading on our manuscript and we have polished the language to make it readable.

P2P_Figure 2. Purification and EM examination of MuV P-L complex from insect cells

Some points that should be addressed:

Line 30 "Structural analysis indicates that C-terminal tail of MuV nucleoprotein hampers the assembly of helical nucleocapsids" clearly this is not the case, it does not hamper NC assembly in authentic virus infections.

Re:

Thanks for pointing out our improper description.

We have changed this improper description as “*Structural analysis on these in vitro assemblies indicates that C-terminal tail of MuV nucleoprotein negatively regulates the assembly of helical nucleocapsids*” (see Line 30-32).

Thanks.

Line 83 – dense and hyperdense states are not defined here or in the previous literature.

Re:

Thanks so much for pointing out this issue.

In MeV and SeV, several nucleocapsid segments are connected with thin and loosed loops, packing antiparallely inside the virions, as revealed by cryo-Electron Tomography (*Loney et al. J Virol 2009; Liljeroos et al. PNAS 2011*). Comparing to thin and loosed loops, we depicted the straight helical nucleocapsid with the helical pitch between 5.0 and 6.8 nm as the dense state. In MuV, there are two kinds of helical filaments in different pitches based on the 2D classification and 3D reconstruction. MuV nucleocapsids with the larger helical pitch at ~5.6 nm is named as NC_{helix-dense}. Accordingly, MuV nucleocapsids with the smaller helical pitch at ~4.5 nm is hyperdense and named as NC_{helix-hyper}.

We have followed the reviewer’s suggestion and defined the “dense state” in the modified introduction (see Line 47-49; Line 204-206).

Thanks.

118 – text here is obscure – needs to be clear that forms discussed are a consequence of heterologous expression.

Re:

Thanks for pointing out this issue.

We followed the reviewer’s advice and emphasized the expression of MuV nucleoproteins are heterologous. The added sentence is “*Herein, we performed heterologous expression on MuV nucleoprotein, used cryo-Electron Microscopy (cryo-EM) as the major approach and resolved 5 high-resolution MuV N-RNA assemblies, including 2 ring-like structures in 13 and*

14 protomers, 1 stacked-ring filament and 2 nucleocapsids with distinct helical pitches” (see Line 75-77).

Thanks.

180 – Ring structures were solved using tilted data at 20 and 40 degrees – the map looks poor in supplemental figure 3 – what was the angular distribution in these reconstructions, as there is still a large missing cone after tilting? Is there sufficient resolution in z? Is the resolution measured isotropic?

Re:

Thanks for the great question.

Due to the flat shapes, MuV N-RNA rings only show top-on views in holey carbon grids or on thin carbon coated holey carbon grids. To deal with the preferred orientation, we collected tilted data at 20° and 40° for side views. Due to the strong preferred orientation, the missing cone still exists in our reconstruction as revealed by angular distribution (see in *P2P_Figure 3*), which makes the EM density in z direction partially stretched. The final reconstruction maps look not as good as the helical reconstructs. We followed the reviewer’s advice and measured 3D FSC as described (*Yong Zi Tan et al. Nature Methods, 2017*). The 3D FSC shows that x, y directions have the resolution at ~3.1 Å while z direction has a relatively poor resolution at 4.6 Å. We have to say that the resolution in x, y and z directions are not isotropic. We have updated 3D FSC in *Supplemental Figure 1b*. Meanwhile, we also revised the methods in the revised manuscript.

Besides MuV N-RNA rings, we also obtained the stacked ring structures and helical nucleocapsids. Among all structures we resolved, NC_{helix-Δarm} has the highest resolution at 2.9 Å with the best EM map in quality. We built an accurate atomic model from NC_{helix-Δarm} and utilized this model to polish the atomic models for other structures including MuV N-RNA ring structure as we mentioned in *Line 215-217*.

Thanks.

P2P_Figure 3. Angular distribution and 3D FSC of MuV N_{ring-13p}

a. Angular distribution of MuV N_{ring-13p} reconstructed from combined datasets at 0°, 20° and 40°. Peaks at 0°, 20° and 40° can be recognized from the distribution map, which indicates the right alignment during 3D reconstruction. Unfortunately, there are almost no any particles at the higher tilted angles, pointing to serious missing wedge even after tilting.

b. Angular distribution of MuV N_{ring-13p} reconstructed only from dataset at 40°. Following the protocol from Lymukis's lab (Yong *et al.* @Lymukis, *Nature Methods* 2017), a little bit better reconstruction relative to combined dataset as in (a) is yielded. The angular distribution is limited to 40°.

c. 3D FSC curve of MuV N_{ring-13p}.

180 – SDS pages shows some degradation – are they sure that the C-arm is present?

Re:

Thanks for the good question.

As shown in *P2P_Figure 4*, the predicted molecular weights for full-length MuV nucleoprotein, $N_{\Delta\text{tail}}$ and $N_{\Delta\text{arm}}$ are 64, 49, and 42 kDa, respectively. After placing full-length MuV nucleoproteins under 4 °C for 4 weeks, cleavage via residual impurities occurs on nucleoproteins, and smear bands from 43 to 52 kDa are visible in the SDS page gel. The cut protein bands are obviously larger than $N_{\Delta\text{arm}}$ with the molecular weight of 42 kDa, which is the first evidence to verify that nucleoproteins still keep their C-arms.

Meanwhile, after high-resolution reconstruction, C-arm is resolved with high resolution in $NC_{\text{helix-dense}}$, which is the other evidence to verify the existence of C-arm after slow degradation.

Thanks.

P2P_Figure 4. Angular distribution of MuV $N_{\text{ring-13p}}$

Bands from Fig. 1a, Fig. 3a, and Fig. 5e were subjected for semi-quantitative analysis and the relative amount in each SDS-Page gel was normalized. The molecular weights for $NC_{\Delta\text{C-tail}}$ and $NC_{\Delta\text{C-arm}}$ were marked in dashed lines.

188 - *"Thus, the flexible C-arm of MuV $N_{\text{ring-13p}}$ and $N_{\text{ring-14p}}$ will not get involved in nucleoprotein oligomerization via forming a stable interface with the $\alpha 16$ helix from the neighboring protomer"... as has been shown for...(another virus?).*

Re:

Thanks for the great suggestion.

We have modified the sentence to *"Thus, the flexible C-arm of MuV $N_{\text{ring-13p}}$ or $N_{\text{ring-14p}}$ will not form a stable interface with the $\alpha 16$ helix from the neighboring protomer, and might not play an essential role in nucleoprotein oligomerization as in MeV, NDV and PIV"* (see Line 113-115).

Thanks.

189-200- *I found this text quite hard to follow and the legend of figure 1 was similarly challenging (and the figures very small).*

Re:

Thanks for the great question.

We followed the advice and split the *Figure 1* into two new figures. Meanwhile, we polished both text and figure legends.

Thanks.

1244. – How can the hyper-dense form help with packing – firstly it is rigid not flexible, secondly it can no longer function as a template without Ntail. The hyper-dense form solved cannot be a biologically relevant structure as it lacks critical regions of the N protein.

Re:

Thanks for your great question.

Previous cryo-ET on MeV and SeV virions show that nucleocapsids are separated into several straight segments, which are connected with thin and loosed loops, as shown in *P2P_Figure 5*. The straight nucleocapsids are the major components in the virions, similar to the recombinant nucleocapsids *in vitro*. Our high-resolution structures on recombinant MuV nucleoproteins show that NC_{helix-dense} and NC_{helix-hyper} co-exist, which is similar to the observations as in MeV (*Bhella et al. J Mol Bio 2004*). Very interestingly, in both MuV and MeV, helical nucleocapsids in different helical pitches are compatible. In some regions, hyperdense states can provide structural plasticity for nucleocapsids to enwrap into the virion. Based on these, we proposed that hyperdense nucleocapsids help with packing.

We agreed with the reviewer that NC_{helix-hyper} with the C-arm cleaved are most likely not to exist in virions, due to the absence of N-tail and C-arm. Our speculation is that C-arm might be re-positioned by Phosphoproteins or others to form partial hyperdense nucleocapsids.

We have toned down the voice about the structural compatibility between MuV NC_{helix-dense} and NC_{helix-hyper}.

Thanks.

P2P_Figure 5. Cryo-ET on SeV and MeV virions. Straight nucleocapsids can be distinguished.

1251 – protection against nuclease digestions in the absence of C-terminal sequences of N is not biologically relevant - as the helix does not exist in this state in authentic viral infections (or at least has not been demonstrated to)

Re:

Thanks for the great comments.

One of the key roles of nucleocapsids is to protect RNA genome from degradation. No matter in MuV N-RNA ring-like structures or nucleocapsids, RNA strands are visible in RNA gel. Comparing to N-RNA ring-like structures or NC_{helix-dense}, the protection of NC_{helix-hyper} against nuclease digestion is more impressive that ~80 nucleotide band is still visible in the RNA gel after the incubation at 4 °C for 180 days. Structurally, the space between two protomers in neighboring rungs in NC_{helix-dense} will be shrunk from ~8,685 Å³ to ~7,279 Å³ in NC_{helix-hyper}, making it even harder for any nucleases to access the embedded RNA.

Just as what the reviewer mentioned, NC_{helix-hyper} caused by the missing of C-arm should not be biologically relevant. Considering that C-arm is not as essential as N-arm in the assembly of helical nucleocapsid (*please refer to the reply to reviewer-2*), C-arm might be re-positioned by phosphoprotein or other proteins. We are resolving the N-P-L supercomplex and hope to address this question in the near future.

We have followed the advice and toned down our voice on this part.

Thanks.

Supplemental figures do not appear to be revised or to show the analysis of layer lines in power spectra requested by reviewer 3

Re:

Sorry for the confusion.

In our previous revision, we have analyzed the layer lines in power spectra in *supplementary figure 6A* (see in *P2P_Figure 6*).

P2P_Figure 6. Old version for 2D class average images of MuV NC_{helix-dense}, NC_{helix-hyper} and N_{ring-stacked} and their diffraction patterns

To make it clearer, we remodified supplementary figure 6A and attached partial *supplementary Figure 6A* here (see in *P2P_Figure 7*).

P2P_Figure 7. New version for 2D class average images of MuV NC_{helix-dense}, NC_{helix-hyper} and N_{ring-stacked} and their diffraction patterns

Reviewers' comments:

Reviewer #2 (Remarks to the Author):

This is again probably an improved draft, and some of the errors have been corrected. However, where there is new text, yet more spelling and grammatical errors are now present. Two in the Abstract...

The bigger problem is that the authors are still implying much more biological relevance than can safely be interpreted.

In addition, many of the constructive suggestions from each of the referees has been side stepped. That is not a sensible strategy.

I'm afraid that I still cannot support the manuscript in its current form.

Reviewer #3 (Remarks to the Author):

The authors have revised the manuscript for the second time addressing a variety of concerns expressed by the reviewers. Now the manuscript includes comparison to other structures. The biological relevance of these assemblies is still not clear. The manuscript puts forward a possible hypothesis for nucleocapsid plasticity which might be incorrect. However the quality of data presented and the high-resolution structures obtained still make the manuscript impactful.

Reviewers' comments:

Reviewer #2 (Remarks to the Author):

This is again probably an improved draft, and some of the errors have been corrected. However, where there is new text, yet more spelling and grammatical errors are now present. Two in the Abstract... The bigger problem is that the authors are still implying much more biological relevance than can safely be interpreted. In addition, many of the constructive suggestions from each of the referees has been side stepped. That is not a sensible strategy. I'm afraid that I still cannot support the manuscript in its current form.

Re:

Thanks again for reviewing our manuscript.

We are so sorry about the spelling and grammatical errors in the manuscript. As suggested by the reviewers and the editor, we have polished the language via Accdon LetPub. We believe the new manuscript is more readable and much clearer than before.

More importantly, we have toned down our voice on the biological relevance of the resolved high-resolution MuV nucleocapsids. Briefly, we would not link the structural plasticity with the genome protection and we had removed the improper statement about the structural transition from ring-like structure to helical nucleocapsids. To better summarize our high-resolution structures of MuV nucleocapsids and avoid possible overinterpretation, we also changed the title to “Cryo-EM structures of mumps virus nucleocapsids and the structural plasticity”.

Thanks.

Reviewer #3 (Remarks to the Author):

The authors have revised the manuscript for the second time addressing a variety of concerns expressed by the reviewers. Now the manuscript includes comparison to other structures. The biological relevance of these assemblies is still not clear. The manuscript puts forward a possible hypothesis for nucleocapsid plasticity which might be incorrect. However the quality of data presented and the high-resolution structures obtained still make the manuscript impactful.

Re:

Thanks so much for reviewing our manuscript.

MuV is a highly contagious pathogen, and unfortunately, the high-resolution structures for MuV nucleoproteins are still missing. We resolved several high-resolution structures of MuV rings and helical nucleocapsids, which is helpful to understand the molecular mechanism for MuV nucleoproteins to prevent assembling into rings and the structural plasticity inherent in MuV nucleocapsids.

Even though the structures are interestingly, we totally agree with the reviewers that the biological significance for the structural plasticity is still unclear. Our bold hypothesis is that structural plasticity might be relevant to genome condensation. To avoid overinterpretation or misleading to readers, we have revised the title of the manuscript to “Cryo-EM structures of mumps virus nucleocapsids and the structural plasticity” and removed the improper description on the transition between MuV rings and helical nucleocapsids.

Thanks.